# Maternal Folic Acid Supplementation Differently Affects the Small Intestinal Phenotype and Gene Expression of Newborn Lambs from Differing Litter Sizes

**DOI:** 10.3390/ani10112183

**Published:** 2020-11-22

**Authors:** Zhen Li, Bo Wang, Heqiong Li, Luyang Jian, Hailing Luo, Bing Wang, Can Zhang, Xingang Zhao, Ying Xue, Sijia Peng, Shuxian Zuo

**Affiliations:** State Key Laboratory of Animal Nutrition, College of Animal Science and Technology, China Agricultural University, Beijing 100193, China; lizhen6394@126.com (Z.L.); wangboforehead@163.com (B.W.); Joan374261795@163.com (H.L.); jianluyang@cau.edu.cn (L.J.); wangb@cau.edu.cn (B.W.); zhang_can@cau.edu.cn (C.Z.); 1404010216@cau.edu.cn (X.Z.); 18801586516@163.com (Y.X.); scarlett_sjj@163.com (S.P.); zuosx5029@163.com (S.Z.)

**Keywords:** folic acid, litter size, maternal nutrition, small intestinal development, morphology, gene expression, lambs

## Abstract

**Simple Summary:**

Maternal folic acid (FA) level during gestation is a factor affecting fetal development due to the vital role of FA in nucleotide synthesis. As a key organ of lamb, the development of the small intestine during gestation in response to maternal FA is not clear. Therefore, the current study evaluated the intestinal phenotype and development-, apoptosis- and digestion-related genes expression in newborn lambs whose mothers supplemented FA during gestation. FA supplementation improved the ratio of small intestinal weight to live body weight, intestinal muscle layer thickness and the expression of *IGF-I*, *BCL-2* and *SGLT1*. Meanwhile, triplets indicated poorer morphological development compared to twins, but the expression of several genes was higher in twins compared to triplets. Overall, maternal FA supplementation during gestation increased the development of offspring’s small intestinal.

**Abstract:**

The purpose of this study was to investigate the effect of maternal dietary folic acid (FA) supplementation during gestation on small intestinal development of newborn lambs of different litter sizes, focusing on the intestinal morphology and development-, apoptosis- and digestion-related genes expression. One hundred and twenty Hu ewes (*Ovis aries*) were inseminated and randomly allotted to three groups. One group received a control diet [without FA supplementation, control (CON)] and the other two groups received control diets supplemented with different amount of FA [16 or 32 mg FA per kg dry matter (DM), i.e., F16 and F32] during pregnancy. After lambing, according to the dietary FA levels and litter size (twins, TW; triplets, TR), the neonatal lambs were divided into 6 (TW-CON, TW-F16, TW-F32, TR-CON, TR-F16, TR-F32) treatment groups. The results showed that the ratio of small intestinal weight to live body weight and the thickness of the intestinal muscle layer in the offspring was enhanced significantly with increasing maternal FA supplementation (*p* < 0.05). Meanwhile, the expression levels of insulin-like growth factor I (*IGF-I*), B-cell lymphoma-2 (*BCL-2)* and sodium/glucose co-transporter-1 (*SGLT1*) in the small intestines of the newborn lambs were increased, while the opposite was true for Bcl2-associated × (*BAX)* in response to FA supplementation (*p* < 0.05). Moreover, the small intestinal weights of twins were significantly higher than those of triplets (*p* < 0.01), and the expression levels of *IGF-I* (*p* < 0.05), sucrase-isomaltase (*SI)* (*p* < 0.05) and solute carrier family 2 member 5 (*SLC2A5*) (*p* < 0.01) were significantly lower than those in triplets. These findings suggest that maternal FA supplementation could improve the offspring’s small intestinal phenotype and the expression of development-, apoptosis- and digestion-related genes, so it could promote the small intestinal development of newborn lambs. Furthermore, the small intestine phenotypic development of twins was generally better than that of triplets, while the expression levels of the above genes of twins were lower than those of triplets.

## 1. Introduction

The development of an animal is the result of the interaction between its genetic potential and environmental conditions [1]. The intrauterine environment is a determinant factor of fetal growth, which depends on the transfer of nutrients through the maternal placenta [2,3]. Folic acid is one of the water-soluble B vitamins and performs critical functions in amino acid metabolism, DNA synthesis and repair, as well as gene expression and modification [4]. Folic acid is important during gestation because it affects the embryonic formation and fetal development by promoting the synthesis of DNA, RNA and various proteins [5]. It was shown that FA supplementation in pregnant rats could increase the offspring’s birth weight [6]. Furthermore, the supplementation with methyl donors such as FA during gestation could improve the growth performance, carcass traits and meat quality of the piglets [7]. Therefore, FA in the maternal diet has important implications for fetal growth and its organ development. The small intestine, as an important digestive organ, is essential for the growth and health of animals. The ruminant offspring intestinal tissues are responsive to maternal nutrition during gestation [8]. As an important precursor of nucleotide synthesis, the FA level influences the growth of various tissues in the gestation, especially for the small intestine, an organ that develops rapidly in mid and late pregnancy. According to early studies, FA deficiency in the maternal diet led to impaired intestinal development of fetal mice [9] and also affected small intestinal differentiation and barrier function in rats [10].

Rumen microbes in ruminants can synthesize FA. However, in some physiological stages, the synthesis cannot meet the increased needs of animals to maintain high level production, such as the lactation period of high-producing dairy cows [11,12]. A previous study also reported that the FA supply from diet and microorganisms in ewes may not satisfactorily meet the requirements of both the mother’s metabolism and fetal growth in the gestation period, especially for the high reproductive breeds [13]. As a famous breed with a large litter size in China, Hu sheep generally produce twins and triples, which can be used as an excellent animal model to investigate FA supplementation during pregnancy [14]. Our previous study has found that the supplementation of FA in pregnancy could increase the offspring’s birth weight, and the numerical increase in birth weight was greater for triplets than twins [15]. Additionally, in lambs that were fed to 6 months of age, it was found that the maternal FA supplementation contributed to improvement of the offspring’s production performance by increasing intestinal enzyme activity and total digestibility [16]. Moreover, an increase in birth weight is often accompanied by an increase in organ weight and small intestinal development originates from the fetus. Therefore, we proposed the following hypothesis: maternal supplementation with FA may regulate the development of the fetus, including that of the small intestine, and thus it may affect postnatal performance. This experiment was conducted to investigate the effect of maternal FA supplementation during gestation on the newborn lambs’ intestinal development of differing litter sizes.

## 2. Materials and Methods

### 2.1. Animals and Diets

This experiment was approved by the Institutional Animal Care and Use Committee of the China Agricultural University (Permit number: DK996). One hundred and twenty Hu sheep with signs of estrus, twice delivery, similar weight and health status were selected. After artificial insemination, they were randomly divided into three groups with forty ewes in each group. The artificial insemination was performed using the same batch of semen and was completed on the same day. The three treatments were supplemented with 0 (CON), 16 (F16) or 32 (F32) mg/kg (DM basis) of rumen-protected FA into the control diet until parturition. All the animals were raised in individual pens (size: 1.5 × 3 m). Twenty-eight days after artificial insemination, pregnancy was checked using type-B ultrasonography, and nonpregnant ewes were removed from the groups. Thirty-five ewes without pregnancy were eliminated throughout the trial period. After lambing, according to their litter size (twins, TW; triplets, TR) and supplemental level of FA (CON, F16 and F32) were divided into six treatment groups in a 2 × 3 factorial design (TW-CON, TW-F16, TW-F32, TR-CON, TR-F16 and TR-F32). 

In this study, the purity, rumen passing rate and small intestinal absorption rate of FA were 99.8%, 92.60% and 85.59%, respectively. The control diet was formulated according to the National Research Council 2007 (NRC 2007) [17]. The adequate dietary ingredients and nutrient levels during the early (from mating to day 90) and late (from day 91 to lambing) gestation were provided to all animals. Folic acid was added to the total mixed ration (TMR) of the ewes and fed twice every day at 8:00 and 18:00. Clean water was available for the animals all the time. The components and nutrient levels of the basal diet are shown in Table 1.

### 2.2. Data and Sample Collection

After the birth of lambs, the litter size of all ewes was recorded, and only twins and triplets were selected for this experiment, of which 66 ewes met the requirements for lambing twins and triplets. The number of ewes that gave birth to twins and triplets were 16 and 5, 13 and 13, and 10 and 9 for the CON, F16 and F32 groups, respectively. The number of twins and triplets used in this experiment were 15 and 6, 11 and 11, and 8 and 14 for the CON, F16 and F32 groups, respectively. DMI was similar throughout the entire gestation period (both early and late stages) in all treatment groups (*p* > 0.05). There was no significant difference between the initial weight of ewes giving birth to twins (44.00 ± 0.53 kg) and triplets (44.20 ± 0.57 kg; *p* > 0.05). Maternal FA supplementation had no effect on the average postpartum weight of ewes, but the postpartum weight of ewes giving birth to triplets was significantly lower than that of ewes giving birth to twins [15].

After lambing, sixty-five twins and triplets with near-average birth weights were euthanized after birth by carbon dioxide indrawing and then followed by exsanguination. After the opening of abdominal cavity and isolation of the small intestine, it was rinsed with normal saline and weighed. The proximal portion of the duodenum and jejunum (a piece approximately 2 cm long) were then removed, split gently using a scalpel, rinsed clean with precooled physiological saline and fixed in 4% paraformaldehyde solution, replacing the fixative after 24 h. Subsequently, approximately 2 cm of anterior duodenum and jejunum samples from each animal were collected in RNase-free cryopreservation tubes and stored in liquid nitrogen for qRT-PCR test.

### 2.3. Small Intestinal Morphology

The anterior segment specimens stored in 4% paraformaldehyde solution were paraffin-embedded and sectionalized, and intestinal morphological analysis was performed with eosin and hematoxylin staining (*n* = 6 for each group). Villous heights, thickness of intestinal muscle layer and crypt depths of intestine for each lamb were measured at intact and well-oriented villi using scanned images of six separate areas obtained from the front part of the duodenum and jejunum were examined at 400× magnification. The measurements were done by means of an imaging software (Panoramic Viewer, 3DHISTECH Ltd., Budapest, Hungary).

### 2.4. qRT-PCR

Total RNA was extracted from the duodenal and jejunal section of the newborn lambs (*n* = 6 for each treatment) using an RNA extraction kit (Tiangen, Biochemical Technology Co., Ltd, Beijing, China). The extracted RNA was dissolved in RNase-Free ddH_2_O. A Nanodrop spectrophotometer (Implen, Inc., Westlake Village, CA, USA) was used to measure the purity of RNA (OD260/280 ratio). First, strand cDNA was synthesized from the RNA through reverse transcription using FastQuant DNA first-chain synthesis kit (Tiangen, Biochemical Technology Co., Ltd, Beijing, China). The reaction was carried out in a total volume of 20 μL containing 2 μL of 10× Fast RT Buffer, 1μL of RT Enzyme Mix, 2 μL of FQ-RT Primer Mix, 2 μL 5× gDNA Buffer, 8 μL of total RNA and 5 mL of RNase-free ddH_2_O. The qRT-PCR was performed on an iQ5 system (Bio-Rad, Hercules, CA, USA) by using the 2× SYBR Green Qpcr Mix kit (Tiangen, Biochemical Technology Co., Ltd, Beijing, China), which was aimed to analyze the expression of development- (*IGF-I*, *GLP2R* and *EGF*), apoptosis- (*BCL-2* and *BAX*) and digestion-related (*MGAM*, *PEPT1*, *SLC2A2*, *LCT*, *SGLT1*, *SI* and *SLC2A5*) genes in the front duodenal and jejunal segment of newborn lambs [18,19,20,21,22,23,24]. The 25 μL Real-Time PCR reaction mix contained 12.5 μL 2× SuperReal Premix Plus, 0.5 μL Forward Primer (10 μM), 0.5 μL, Reverse Primer (10 μM), 1 μL cDNA and 10.5 μL RNase-free ddH_2_O. The design and synthesis of primers were made by Shanghai Generay Biotech Co., Ltd (Shanghai, China). The sequences of the forward and reverse primers are shown in Table 2, and the housekeeping gene *GAPDH* was used to calculate the relative expression levels of each target gene in the two tissues. The qRT-PCR used the value of the threshold cycle (Ct) for the relative quantification of gene amplification. The expression level of the target gene relative to those of *GAPDH* gene was quantified using the Ct value comparison method of the 2^−^^∆∆Ct^ method.

### 2.5. Statistical Analysis

The data of weight and morphology of the small intestine and genes expression were analyzed using the generalized linear model (GLM) of statistical package SPSS version 23.0 (SPSS, IBM, Inc., Chicago, IL, USA) to assess the effects of litter size (TW and TR) and dietary FA supplementation levels (CON, F16 and F32). Treatment means were compared using Duncan’s new multiple range test with a significance level of *p* < 0.05. Polynomial analysis was conducted to test the linear or quadratic response to the levels of FA supplementation in the diet at the 0.05 level. The variables in different litter sizes were compared using the independent samples *T*-test at the 0.05 level. Results are shown as mean ± SEM. The relationships between genes and phenotypes for the various treatments were determined using correlation analysis.

## 3. Results

### 3.1. Weight and Morphology of the Small Intestine

The effect of litter size and FA supplementation on intestinal morphology is shown in Table 3. The small intestinal weight of twin lambs was significantly higher than that of the triplet lambs (*p* = 0.001). In addition, the ratio of small intestine weight to live body weight increased linearly with the FA supply in maternal diet (*p* = 0.005).

The results and image of the morphology of duodenal segments are presented in Figure 1, Figure 2 and Figure 3, respectively. There was a significant (*p* = 0.011) effect of the interaction between litter size and FA supplementation on duodenal muscle layer thickness of the lambs. The thickness of duodenal muscle layer increased linearly with maternal dietary FA supplementation in the twin-born lambs (*p* < 0.001), while FA supplementation had no influence on the duodenal muscle layer thickness among the triplet-born lambs (*p* = 0.134). The FA concentration and litter size had significant effects on jejunal muscle layer thickness. The jejunal muscle layer thickness increased linearly with the increase of FA level (*p* = 0.019). Meanwhile, jejunal muscle layer thickness of triplets was significantly higher than twins (*p* = 0.019). The villus height and crypt depth of duodenum and jejunum tissue were similar among the treatment groups (*p* > 0.05).

### 3.2. Gene Expression

The effect of litter size and FA supplementation on intestinal development-related genes expression is shown in Table 4. The interaction between FA supplementation and litter size had a significant effect on the expression of *IGF-I* in the duodenum (*p* = 0.005). The duodenal expression of *IGF-I* increased linearly with dietary FA supplementation in triplet-born lambs (*p* = 0.039), but it was not affected by FA supplementation in twin-born lambs (*p* = 0.310). Furthermore, the expression of *IGF-I* in the jejuna of twins was remarkably lower than triplets with no significant interaction effect between litter size and FA supplementation (*p* = 0.041). However, expression of *EGF* and *GLP2R* genes in the small intestine showed no statistical differences among the treatment groups (*p* > 0.05).

Results of apoptosis-related genes expression are shown in Figure 4. In the duodenum, maternal diet FA supplementation significantly linearly increased the expression of *BCL-2* (*p* = 0.027). For the jejunum, the expression of *BAX* in newborn lambs was influenced by the interaction effect between litter size and FA supplementation (*p* = 0.046), which showed a quadratic response to FA supplementation in twin lambs (*p* = 0.034).

Figure 5 shows the results of small intestinal digestion-related genes expression of newborn lambs. The interaction between litter size and FA supplementation had a significant (*p* < 0.05) effect on the expression of *SGLT1* in the duodenal tissue and on the expression of *LCT* and *SLC2A5* in the jejunal tissue. The expression level of *SGLT1* increased linearly with the supply of FA in triplet lambs (*p* = 0.005), while the expression level of *SLC2A5* varied quadratically with the FA supplementation in triplets (*p* = 0.021). Meanwhile, the addition of FA had no effect on the expression of *SGLT1* in the duodenal tissue and *SLC2A5* in the jejunal tissue in twin-born lambs (*p* > 0.05). The expression level of *LCT* was not affected by FA supplementation in triplets (*p* = 0.511), but it showed a quadratic response to FA supplementation in twins (*p* = 0.005). Moreover, the effect of litter size on the expression of *SI* and *SLC2A5* was also significant in the duodenal tissue. The expression of *SI* and *SLC2A5* were elevated in the triplets compared to those in the twins (*p* < 0.05). Results of other genes (*MGAM*, *PEPT1*, *SLC2A2*) with no significant differences in duodenum and jejunum were not shown.

### 3.3. Pearson Correlation Analysis

We also determined the correlations between gene expression and phenotype parameters (villus height, thickness of muscle layer and crypt depth) of the duodenal and jejunal tissues; the results from the correlation analyses are listed in Table 5 and Table 6. In the duodena, the expression levels of *MGAM*, *LCT*, *SGLT1*, *SI* and *SLC2A5* were significantly positively correlated with crypt depth (*p* < 0.05). Similarly, the expression of *IGF1* had a positive correlation with jejunal crypt depth (*p* = 0.027). In addition, positive correlations were found between *MGAM* and villus height (*p* = 0.029), as well as between *PEPT1* and muscle layer thickness in jejunum (*p* = 0.008). In general, although there were no significant correlations between thickness of muscle layer and gene expression in the duodenum tissue, there was a general trend of negative correlation. In addition, villus height seems to be a nonsignificant positive correlation with gene expression in the duodenum and jejunum tissue.

## 4. Discussion

As a vital methyl donor, FA plays a role in the maintenance and development of the fetus during gestation [25]. Maternal supplementation of methyl donors such as FA has been reported to decrease incidence of pregnancy failure and increase litter size in early parity sows [26]. Unlike the monogastric animals, FA can be synthesized in the rumen of ruminants, so according to the classic ruminant nutrition theory, there is no need for dietary FA given in ruminants [27]. However, the supply of FA at a particular physiological stage is of great importance to the assurance of production performance due to the variation of nutrient requirements at different physiological stages of ruminants. For example, a previous study reported a high demand for FA in the tissues of lactating and gestating cows [28]. The functional maturity of the gastrointestinal system is critical to the survival of the neonate [29]. Insufficient maternal nutrition has a deleterious effect on the small intestine of the offspring. These effects generally include a reduction in the weight and length of the small intestine, resulting in a decrease in the functional area and a perturbed development of the small intestine [30,31]. The weight of small intestine and its proportion of body weight are important indicators of its development. In the present study, it was shown that the increase of the small intestine/live body weight ratio in FA-supplemented groups was the most direct indicator of improved intestinal development because of maternal dietary FA supplementation.

During the morphological assessment of the intestinal tract in livestock, the villus height, crypt depth and muscle layer thickness of the small intestine are important indicators that reflect the ability of gastrointestinal tract to digest and absorb nutrients [32]. The intestinal phenotype of the offspring could be affected by FA concentration in the mother’s diet. A previous study has shown that the deficiency of methyl donors at the time of gestation and lactation resulted in increased apoptosis in the crypts and decreased differentiation of villus epithelial cells in 26-day-old rats [10]. Sufficient FA is vital for maintaining the normal structure of the human small intestine [33]. In our experiment, similar results were obtained; supplementation of the ewe’s diet with FA during pregnancy markedly improved muscle layer thickness in offspring, suggesting an improvement in intestinal function. Furthermore, it is generally accepted that litter size is also an important factor affecting the development of the offspring [34]. Depending on the number of fetuses present in the uterus, the same maternal nutritional level provides different amount of nutrients for tissue growth, which can lead to obvious discrepancy in fetal development of different litter sizes [31]. Animals with nutritional limitations in the fetus tend to have poorer morphological development of small intestine after birth than the normal-nutrition-supplied animals [35]. In this study, the weight of the small intestine of twins was significantly higher than that of triplets, with the development of the small intestine of twin lambs being superior based on phenotypic evaluation—that is to say, under the same nutrition level of the ewes, the triplets will receive less nutrition from the dam than the twins, so deficiency of nutrition leads to apparent morphological defects in small intestinal development. Finally, since most of the effects of F16 and F32 on small intestine morphological development were not significantly different, the optimal level of FA supplementation on intestine development deserves further study.

The results of this trial showed that not only the morphological aspects of development were affected but also that the supply of FA can influence the genes expression level in the small intestine. Interestingly, the expression of *IGF-I* in twin lambs was significantly lower than that in triplet lambs, but the result of the small intestinal phenotype was that twins developed better than triplets. Insulin-like growth factors (IGFs) are important growth factors that regulate the development of the small intestine [36]. Due to the role of IGF-I in promoting division and differentiation of cells, it has been found in pigs [37] and sheep [38] that IGF-I supplementation could improve the morphology of small intestine and promote the development of small intestinal epithelium. Comparing with twins, triplets compete more for maternal nutrition during the fetal stage and are more likely to suffer from developmental defects caused by inadequate nutrition. The disturbance of environmental stress caused by undernutrition on ontogenetic development will reduce the adaptability of animals. However, the fetal programming determined by epigenetic modification can be adjusted by influencing the gene expression, so this genetic change caused by gene plasticity that tends to restore the normal phenotype of the body is called “genetic compensation” [39,40]. Therefore, a possible explanation for the higher expression of *IGF-I* in the jejunal tissue of triplets than twins is that the body compensates for the lower small intestine weight of triplet-born lambs by increasing expression of *IGF-I*. A similar experiment found that the intestinal morphology of offspring piglets with nutritional restriction during gestation was impaired, while FA supplementation increased the expression of antiapoptotic genes in jejunum tissues [41].

Chronic FA deficiency can lead to an imbalance in the defense system against cell antioxidants and increased apoptosis, indicating that FA is crucial in cell cycle regulation [42]. When we examined the expression of intestinal apoptosis-related genes, we found that in the duodenum, FA addition during gestation could prominently improve the expression of antiapoptotic gene *BCL-2* in newborn lambs. There was also a linear decline in the expression of proapoptotic gene *BAX* with the addition of FA. These results were in consistent with previous reports about the influence of maternal FA supplementation on morphology and apoptosis-related gene expression in the jejuna of newborn intrauterine-growth-retarded piglets [41]. Furthermore, *BAX* and *BCL* are important genes that regulate apoptosis, which is crucial for maintenance of normal intestinal morphology [43]. Therefore, combined with the results of intestinal morphology, we suggest that FA supplementation during pregnancy might affect the degree of intestinal development by changing the apoptosis in the intestinal epithelial cells.

The small intestine is an important organ involved in the digestion of nutrients and its function can be used to evaluate its developmental quality [44]. D-glucose mainly relies on two kinds of transporters in intestinal mucosal epithelial cells the sodium/glucose cotransporter (SGLT) family and the glucose transporter (GLUT) family—to enter the tissues of the body, which is the main energy substance of the body. Glucose absorption in the intestinal tract mainly occurs in the anterior segment of the small intestine, so the expression of *SGLT* family genes in the small intestine is extremely important for the ability to absorb glucose [45]. Moreover, the maternal nutrition during gestation could affect the concentration of intestinal enzymes in the offspring [46]. Lactase (LCT), also known as β-galactosidase, is an enzyme, which under normal circumstances catalyzes the hydrolysis of lactose into galactose and glucose. In most mammals, the expression and catalytic activity of *LCT* reached the highest level at birth, decreased significantly after weaning, and the expression in the jejunum was the highest [47]. In this study, dietary FA supplementation in gestation generally increased the expression of *SGLT1* and *LCT*, suggesting that FA may be conducive to lactose hydrolysis and glucose absorption, further supplying energy for the body. In addition, the small intestinal disaccharidase (SI) plays an important role in the utilization of carbohydrates in newborn animals, as disaccharides cannot be directly absorbed and utilized by the small intestine. The *SLC2A5* gene, also known as *GLUT5*, is involved in encoding a fructose transporter [48]. We have found that the genes expression of *SI* and *SLC2A5* significantly increased in triplets compared with that in twins, which is consistent with the result of *IGF-I* expression in jejunum tissue, and consistent with what we found in terms of small-intestine-development-related gene expression. In this case, the increased expression of digestion-related genes in triplets could actually be compensation for the smaller absorptive surface area in the small intestine.

The relationship between small intestinal morphology and gene expression was investigated using a Pearson correlation analysis. Multiple development- and digestion-related genes were positively correlated with crypt depth in the small intestine, which represents the degree of intestinal development because it provides a continuous flow of epithelial cells to the intestinal villi. Taken together, these findings suggested that FA may improve intestinal growth by influencing the expression of some genes that regulate crypt development. We also found that in duodenum and jejunum tissue, most of the digestion-related genes had a nonsignificant negative correlation with the muscle layer thickness. The decrease of the thickness of the muscle layer of the intestine could improve the digestion and absorption of nutrients [49]. In addition, the duodenal villi height was insignificantly positively correlated with the expression of most genes, but not in the jejunum tissue; a possible explanation for this may be due to the tissue specificity of gene expression.

## 5. Conclusions

In conclusion, this study found that FA supplementation during gestation could improve the morphological development of duodena and jejuna, the expression of development-, antiapoptotic- and digestion-related genes, as well as reduce the expression of proapoptotic-related genes in newborn lambs. Meanwhile, litter size affected the growth of the small intestine in offspring as the phenotypic development in twins was better, while the expression level of genes in triplets was higher. The results indicate that FA, to a certain extent (at the level of gene expression), could make up the perturbed development caused by maternal nutritional deficiency in triplet lambs, but not phenotypically. As for which F16 or F32 has the better effect, it cannot be judged based on the existing data and deserve future study.

## Figures and Tables

**Figure 1 animals-10-02183-f001:**
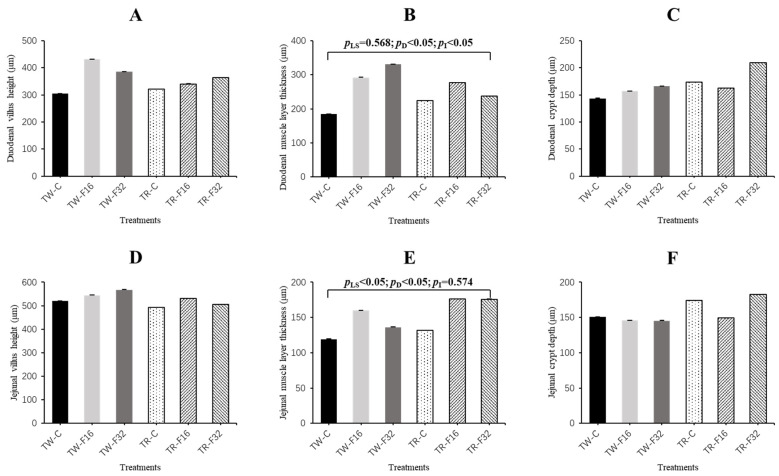
Duodenal villus height (**A**), duodenal muscle layer thickness (**B**), duodenal crypt depth (**C**), jejunal villus height (**D**), jejunal muscle layer thickness (**E**) and jejunal crypt depth (**F**) affected by maternal folic acid supplementation. TW-CON, TW-F16 and TW-F32, twins born from ewes fed 0, 16 or 32 mg·(kg DM)^−1^ folic acid in the control diet, respectively; TR-CON, TR-F16 and TR-F32, triplets born from ewes fed 0, 16 or 32 mg·(kg DM)^−1^ folic acid in the control diet, respectively. *p*_LS_, *p*-value of litter size; *p*_D_, *p*-value of diet folic acid supplementation; *p*_I_, *p*-value of interaction effects between litter size and folic acid supplementation.

**Figure 2 animals-10-02183-f002:**
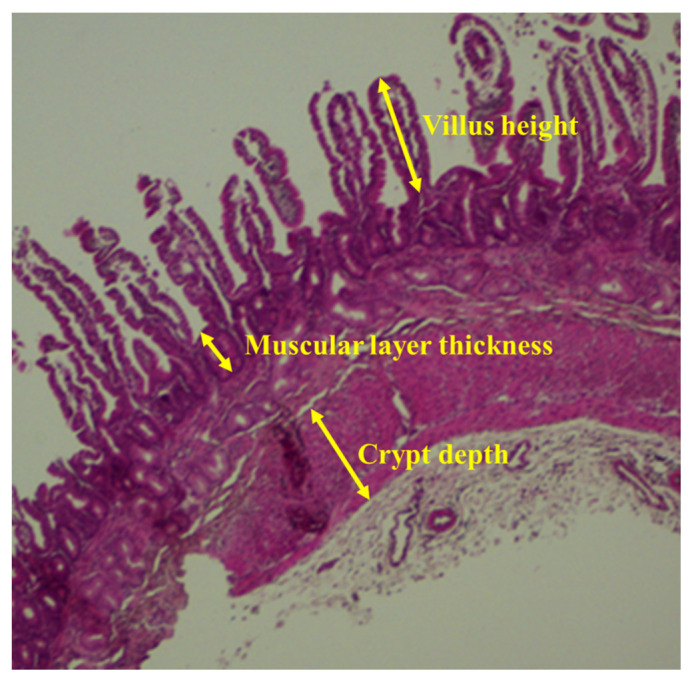
Schematic diagram of small intestinal morphological characteristics.

**Figure 3 animals-10-02183-f003:**
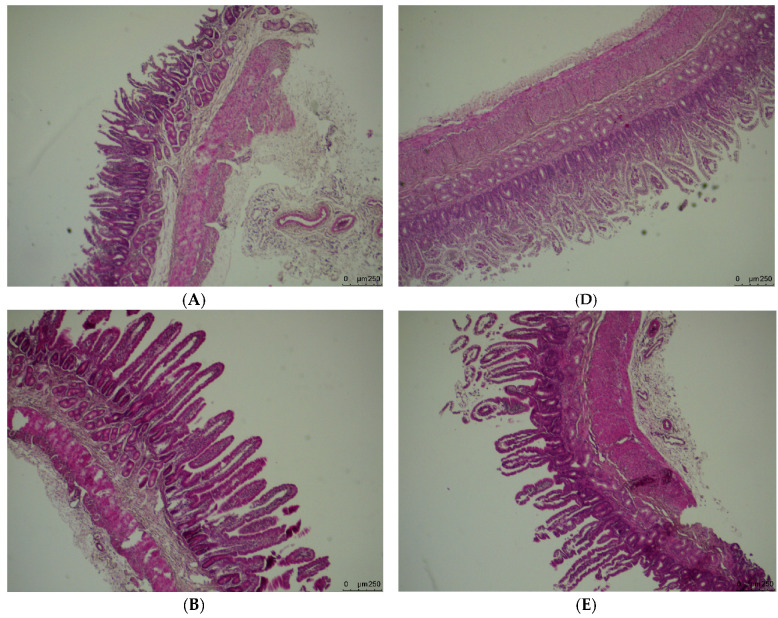
Morphological characteristics of duodenal cross-section of newborn lambs that were fed folic acid by the mother during pregnancy. (**A**–**C**) are cross-sections from twin-born lambs from ewes fed 0, 16 or 32 mg·(kg DM)^−1^ folic acid in the basal diet, respectively; (**D**–**F**) are cross-sections from triplet-born lambs from ewes fed 0, 16 or 32 mg·(kg DM)^−1^ folic acid in the basal diet, respectively.

**Figure 4 animals-10-02183-f004:**
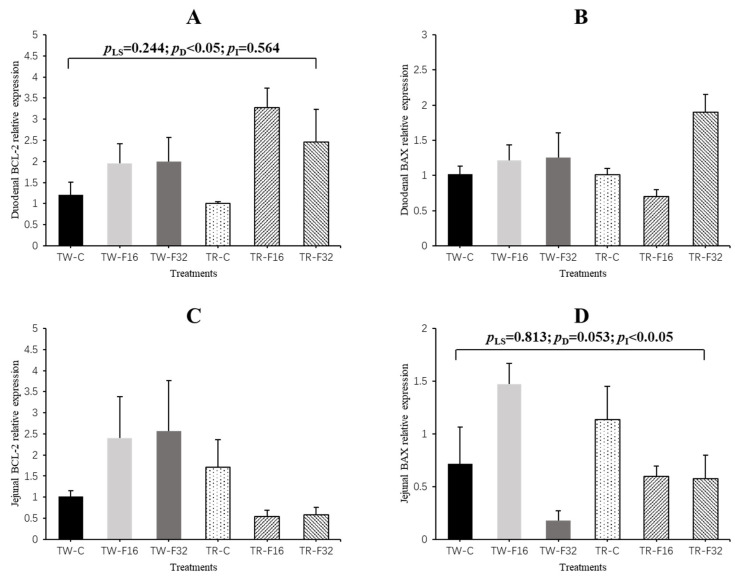
The duodenal gene expression of *BCL-2* (**A**), *BAX* (**B**) and jejunal gene expression of *BCL-2* (**C**) and *BAX* (**D**) affected by maternal folic acid supplementation. TW-CON, TW-F16 and TW-F32, twins born from ewes fed 0, 16 or 32 mg·(kg DM)^−1^ folic acid in the control diet, respectively; TR-CON, TR-F16 and TR-F32, triplets born from ewes fed 0, 16 or 32 mg·(kg DM)^−1^ folic acid in the control diet, respectively. *p*_LS_, *p*-value of litter size; *p*_D_, *p*-value of diet folic acid supplementation; *p*_I_, *p*-value of interaction effects between litter size and folic acid supplementation.

**Figure 5 animals-10-02183-f005:**
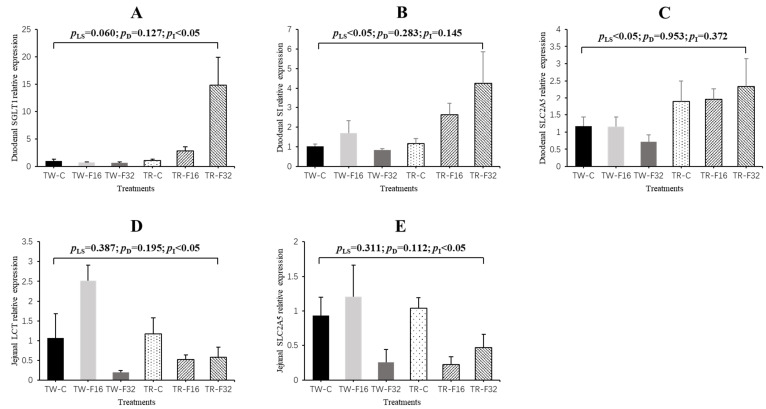
The duodenal gene expression of *SGLT1* (**A**), *SI* (**B**), *SLC2A5* (**C**) and jejunal gene expression of *LCT* (**D**) and *SLC2A5* (**E**) affected by maternal folic acid supplementation. TW-CON, TW-F16 and TW-F32, twins born from ewes fed 0, 16 or 32 mg·(kg DM)^−1^ folic acid in the control diet, respectively; TR-CON, TR-F16 and TR-F32, triplets born from ewes fed 0, 16 or 32 mg·(kg DM)^−1^ folic acid in the control diet, respectively. *p*_LS_, *p*-value of litter size; *p*_D_, *p*-value of diet folic acid supplementation; *p*_I_, *p*-value of interaction effects between litter size and folic acid supplementation.

**Table 1 animals-10-02183-t001:** The composition and nutrient levels in the basal total mixed ration (TMR) (dry-matter basis, %).

Constitution of TMR		Formula of Concentrate			Levels of Nutrients in TMR		
EG		Ingredients	EG	LG	Nutrients	EG	LG
Peanut vines	50.00	Corn	53.00	50.00	DM	92.35	90.96
Whole corn silage	45.00	Soybean meal	9.70	22.50	CP	9.60	10.00
Concentrate	5.00	Rapeseed meal	12.00	7.00	EE	3.51	4.52
Total	100.00	Wheat bran	15.70	11.50	Ash	11.19	8.85
LG		Limestone	1.00	1.00	NDF	49.19	34.15
Peanut vines	27.00	CaHPO4	0.60	0.60	ADF	29.98	24.86
Whole corn silage	28.00	NaHCO3	1.30	1.30	ME ^2)^	1.93	2.15
Concentrate	45.00	NaCl	0.80	0.40	Ca	0.54	0.56
Total	100.00	Vitamin E	0.40	0.10	P	0.37	0.26
		Soybean oil	0.30	0.40			
		Premix ^1)^	5.00	5.00			
		De-mold agent	0.20	0.20			
		Total	100.00	100.00			

TMR, total mixed ration; EG, early gestation; LG, late gestation; DM, dry matter; CP, crude protein; EE, ether extract; NDF, neutral detergent fiber; ADF, acid detergent fiber; ME, metabolizable energy; Ca, calcium; P, phosphorus. ^(1)^ Premix was provided VA 30000 IU, VD 10000 IU, VE100mg, Fe 90 mg, Cu 12.5 mg, Mn 50 mg, Zn 80 mg, Se 0.3 mg, I 0.8 mg and Co 0.5 mg per kg. ^(2)^ ME value was calculated according to the NRC, unit: Mcal/kg.

**Table 2 animals-10-02183-t002:** Primers sequence of fluorescence quantitative gene.

Genes Name		Primers Sequence	Recommended Melting Temperature (℃)	Product Size (bp)
*SGLT1*	Forward	AGAGGTCACAGTTGGAATGGC	60	110
	Reverse	TGATTATGCTCCTGGGGTCTT		
*PEPT1*	Forward	GTAGACGATGGACAGCGACAC	60	211
	Reverse	CAATGAGTTCTGCGAAAGGTT		
*IGF-I*	Forward	CCAGCCTGCTGTTATTTCTTT	60	80
	Reverse	CATTGCGGTTCTGTTGATAGT		
*BAX*	Forward	CTTCCAGATGGTGAGTGAGGC	60	242
	Reverse	GGGTTGTCGCCCTTTTCTACT		
*BCL-2*	Forward	ACTGCTTTCACGAACCTTTTG	60	142
	Reverse	TTGATTTCTCCTGGCTGTCTC		
*SLC2A5*	Forward	ACAAGGATTCCGATGGTGATA	60	113
	Reverse	GCAGGTCTGTCTTCCAATGTC		
*GLP2R*	Forward	CCCCAGCACCCTGTATTCTCC	60	175
	Reverse	ATTACTCGTGGCTGCTCGTCG		
*EGF*	Forward	CAGACAAGTCGCCAGCAAACG	60	195
	Reverse	GCCCTCCCACCTCCTCCAAGT		
*SLC2A2*	Forward	GGCACAAACAAACATTCCACT	60	166
	Reverse	CTCAACCAGCATTTTCCAGAC		
*SI*	Forward	TCAAACCACCGAGCATTAGGG	60	125
	Reverse	AATGAAAAGCCAACCTGGGATA		
*LCT*	Forward	TCTCCAGTGGCGTTGTCTTTC	60	118
	Reverse	TGTCCTCGTCAGCCTATCAGA		
*MGAM*	Forward	GGCCCTAAAACCACATAAAAG	60	162
	Reverse	CTACGGTGTCCACCCCTACTA		
*GAPDH*	Forward	GGCGTGAACCACGAGAAGTA	60	141
	Reverse	GGCGTGGACAGTGGTCATAA		

SGLT1: sodium/glucose co-transporter-1; PEPT1: oligopeptide transporter 1; IGF-I: insulin-like growth factor I; BAX: bcl-2-associated x; BCL-2: b-cell lymphoma-2; SLC2A5: solute carrier family 2 member 5; GLP2R: glucagon-like peptide-2 receptor; EGF: epidermal growth factor; SLC2A2: solute carrier family 2 member 2; SI: sucrase-isomaltase; LCT: lactase; MGAM: maltase-glucoamylase.

**Table 3 animals-10-02183-t003:** Effect of dietary folic acid supplementation during gestation on intestinal index of newborn lambs.

Items	Twins			Triplets			Litter size		Diet			*p*-value		
	CON	F16	F32	CON	F16	F32	Twins	Triplets	CON	F16	F32	Litter Size	Diet	I×
Small intestine weight (g)	118.75 ± 3.09	128.93 ± 5.76	142.00 ± 7.46	106.96 ± 8.33	114.98 ± 3.62	113.01 ± 5.44	127.51 ± 3.22	112.54 ± 3.14	115.39 ± 3.34	121.96 ± 3.65	123.55 ± 5.26	0.001	0.356	0.244
Small intestine/live body weight (%) ^(1)^	3.15 ± 0.07	3.31 ± 0.09	3.48 ± 0.14	3.17 ± 0.17	3.45 ± 0.09	3.49 ± 0.09	3.28 ± 0.06	3.41 ± 0.06	3.15 ± 0.07	3.38 ± 0.07	3.48 ± 0.08	0.122	0.005	0.769

CON, F16 and F32 mean newborn lambs from ewes fed 0, 16 or 32 mg (kg DM)^−1^ folic acid in the basal diet, respectively. I×, interaction effects of litter size and folic acid addition. ^(1)^ The ratio of small intestine weight to live body weight linearly increased response to folic acid supplementation (*p* = 0.005).

**Table 4 animals-10-02183-t004:** Effect of dietary folic acid supplementation during gestation on small intestinal development-related genes expression of newborn lambs.

Items	Genes	Twins			Triplets			Litter Size		Diet			*p*-Value		
		CON	F16	F32	CON	F16	F32	Twins	Triplets	CON	F16	F32	Litter Size	Diet	I×
Duodenum	EGF	1.24 ± 0.36	3.69 ± 0.94	1.58 ± 0.32	1.09 ± 0.23	0.90 ± 0.25	1.08 ± 0.32	2.01 ± 0.46	1.02 ± 0.14	1.08 ± 0.21	3.41 ± 0.65	0.89 ± 0.21	0.055	0.099	0.285
	GLP2R	1.08 ± 0.19	1.00 ± 0.16	0.62 ± 0.13	1.04 ± 0.12	1.00 ± 0.13	1.93 ± 0.47	0.89 ± 0.11	1.28 ± 0.18	0.94 ± 0.11	0.83 ± 0.10	0.91 ± 0.33	0.064	0.675	0.399
	IGF-I ^(1)^	1.07 ± 0.16	0.78 ± 0.12	0.78 ± 0.15	1.08 ± 0.19	2.43 ± 0.66	2.55 ± 0.23	0.88 ± 0.09	1.95 ± 0.30	1.14 ± 0.12	1.46 ± 0.40	1.52 ± 0.31	0.003	0.422	0.005
Jejunum	EGF	0.98 ± 0.35	0.98 ± 0.50	0.80 ± 0.46	1.14 ± 0.35	0.49 ± 0.19	0.45 ± 0.22	1.00 ± 0.23	0.74 ± 0.15	1.07 ± 0.23	0.74 ± 0.21	0.65 ± 0.24	0.352	0.255	0.691
	GLP2R	0.80 ± 0.80	0.60 ± 0.32	1.40 ± 0.81	1.05 ± 0.19	1.24 ± 0.44	1.13 ± 0.11	1.10 ± 0.20	1.12 ± 0.17	0.95 ± 0.21	0.92 ± 0.25	1.29 ± 0.20	0.947	0.729	0.481
	IGF-I	0.69 ± 0.63	0.95 ± 0.20	0.14 ± 0.09	2.02 ± 0.70	0.88 ± 0.17	0.89 ± 0.41	0.60 ± 0.16	1.36 ± 0.29	1.45 ± 0.42	0.92 ± 0.10	0.46 ± 0.28	0.041	0.390	0.360

CON, F16 and F32 mean newborn lambs from ewes fed 0, 16 or 32 mg (kg DM)^−1^ folic acid in the basal diet, respectively. I×, interaction effects of litter size and folic acid addition. ^(1)^ The expression of *IGF-I* increased linearly with dietary folic acid supplementation in the triplets (*p* = 0.039).

**Table 5 animals-10-02183-t005:** Correlation analysis of genes and phenotypes in the duodena of newborn lambs supplemented with folic acid during maternal gestation.

Terms	Villus Height	Thickness of Muscle Layer	Crypt Depth
	*R* (*p*)	*R* (*p*)	*R* (*p*)
BCL-2	0.328 (0.127)	−0.092 (0.660)	0.039 (0.851)
EGF	0.279 (0.197)	0.279 (0.177)	−0.220 (0.280)
MGAM	0.120 (0.577)	−0.226 (0.266)	0.510 (0.007)
PEPT1	0.287 (0.195)	0.161 (0.453)	0.060 (0.785)
SLC2A2	0.012 (0.959)	−0.228 (0.262)	0.147 (0.484)
BAX	0.294 (0.173)	0.124 (0.554)	−0.015 (0.943)
GLP2R	0.337 (0.093)	−0.286 (0.132)	−0.147 (0.448)
IGF1	0.189 (0.367)	−0.145 (0.461)	0.208 (0.288)
LCT	−0.103 (0.639)	−0.039 (0.849)	0.586 (0.002)
SGLT1	−0.032 (0.890)	−0.181 (0.397)	0.646 (0.001)
SI	0.104 (0.630)	−0.130 (0.519)	0.541 (0.004)
SLC2A5	−0.090 (0.683)	−0.153 (0.454)	0.418 (0.034)

*R*: correlation coefficient; *p*: significance (probability). Differences were considered significant at *p* < 0.05.

**Table 6 animals-10-02183-t006:** Correlation analysis of genes and phenotypes in the jejuna of newborn lambs supplemented with folic acid during maternal gestation.

Terms	Villus Height	Thickness of Muscle Layer	Crypt Depth
	*R* (*p*)	*R* (*p*)	*R* (*p*)
BCL	−0.115 (0.593)	−0.304 (0.169)	0.173 (0.441)
EGF	−0.129 (0.556)	−0.193 (0.377)	0.008 (0.969)
MGAM	0.429 (0.029)	−0.181 (0.357)	0.053 (0.787)
PEPT1	−0.058 (0.779)	0.499 (0.008)	0.155 (0.441)
SLC2A2	0.196 (0.338)	0.011 (0.957)	−0.093 (0.645)
BAX	0.085 (0.700)	−0.172 (0.422)	−0.229 (0.283)
GLP2R	−0.189 (0.377)	−0.113 (0.600)	−0.047 (0.827)
IGF1	−0.351 (0.093)	0.112 (0.621)	0.472 (0.027)
LCT	0.083 (0.705)	−0.015 (0.946)	−0.042 (0.852)
SGLT1	−0.360 (0.091)	0.275 (0.227)	0.189 (0.412)
SI	−0.190 (0.408)	0.059 (0.801)	0.178 (0.440)
SLC2A5	−0.430 (0.036)	0.024 (0.915)	0.248 (0.266)

*R*: correlation coefficient; *p*: significance (probability). Differences were considered significant at *p* < 0.05.

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
