# Peer review of "Maternal Folic Acid Supplementation Differently Affects the Small Intestinal Phenotype and Gene Expression of Newborn Lambs from Differing Litter Sizes"

_animals, 2020, doi:10.3390/ani10112183_

Round 1
Reviewer 1 Report
General comments.
This paper is very interesting and is relevant to understanding and improving both human and livestock health. There are sections which require more attention to detail and improved English expression.
Specific comments below.
Simple Abstract
Line 20: you use the term abundance here, and the term expression below. Which is more appropriate? Be consistent?
Suggest improved wording of the last sentence.
Line 21: Compared to twins
Line 22: higher in twin lambs compared to triplets
Abstract
Relies heavily on acronyms which have not yet been defined. Given the abstract is what is found on search engines, it's best to have well defined acronyms here for ease of understanding the content without the need to access the full paper.
Is it necessary to italicise the gene names?
Keywords
These words act as additional searching terms and should differ to the ones in your title. The use of different words enhances the search criteria.
Introduction.
Th information presented here is good and very interesting; however, I strongly recommend rewording the second paragraph for clarification of your arguments. The English expression needs some attention.
Methods
Table 1: the animal number is confusing. 16 twin-bearing ewes, why do you have 15 twin lambs? Why are there not 32 lambs? Same for the triplets. If there 5 triplet-bearing ewes, why were there only 6 triplet lambs? Surely you’d want to pair the lamb data to the ewe.
There is no mention here of the animals actual daily feed intake? Did all the animals consume their daily mixed ration?
As FA was fed throughout gestation based on ewe body weight, how frequently were the ewes weighed throughout your trial? How often did you need to adjust the concentration FA to account for LW changes, and how did you ensure the ewes were receiving the accurate FA concentration throughout gestation?
Given that the triplet-bearing ewes were significantly lighter postpartum, did you monitor the ewe’s body condition score throughout gestation?
Again, the English expression needs some attention. For example. Line 104: Ewes were free access to water? Suggest reword
The statistical analysis is too simple for the results presented. Please provide explicit details of the statistical analysis, including the variables measured.
Results.
If you intend to use the term tendency you should include the F-statistics information to support your claim that these results were in fact tended towards significance. In saying that, a quick change of your significance value from P < 0.050 to P < 0.05 in the statistical analysis section would remove the need to use the term tendency and you could state there was a significant difference.
Line 170: triplet -born. Remove the space before the hyphen.
Line 172: improved is a subjective term. Use increased, was higher, thicker to quantify your results and leave the subjective terms for the discussion.
Figure 1: lacks information about the specific morphological characteristics. Morphological characteristics is vague. Can you be a bit more specific.
The significant correlations were moderate at best and should be stated as such.
You mentioned you conducted a regression analysis yet you have not presented those results. Was there any bias in your data?
Discussion
The discussion is well written with grammatical minor errors.
244: in in. Remove the repeated word
259: switching from FA to folic acid, then back to FA. Be consistent.
Reference
Please cross-check your citation information correctly. There are some author details which have been presented incorrectly.
Author Response
Dear Reviewer,
Thank you for helping us improve the quality of manuscript (ID: animals-963445). We gratefully appreciate the constructive comments made by you. We revised the manuscript thoroughly according to your comments. Any changes of the manuscript were indicated in red in the revised manuscript and listed as follows.
Reviewer 1
Comments and Suggestions for Authors
General comments.
This paper is very interesting and is relevant to understanding and improving both human and livestock health. There are sections which require more attention to detail and improved English expression.
Specific comments below.
Simple Abstract
Line 20: you use the term abundance here, and the term expression below. Which is more appropriate? Be consistent?
Response: We replaced “abundance” by “expression” according to your suggestion in line 21.
Suggest improved wording of the last sentence.
These results indicate that maternal FA might promote the development of small intestine in offspring.
Response: We changed the last sentence to “Overall, maternal FA supplementation during gestation increased the development of offspring's small intestinal” according to your suggestion (line 23-24).
Line 21: Compared to twins
Response: We replaced “than twins” by “compared to twins” according to your suggestion in line 22.
Line 22: higher in twin lambs compared to triplets
Response: We replaced “higher than twin lambs” with “higher in twin lambs compared to triplets” according to your suggestion in line 22-23.
Abstract
Relies heavily on acronyms which have not yet been defined. Given the abstract is what is found on search engines, it's best to have well defined acronyms here for ease of understanding the content without the need to access the full paper.
Response: Thanks very much for your suggestion. We added full name for all genes’ abbreviation, including insulin-like growth factor I (IGF-I), B-cell lymphoma-2 (BCL-2) and sodium/glucose co-transporter-1 (SGLT1), Bcl2-associated x (BAX), sucrase-isomaltase (SI) and solute carrier family 2 member 5 (SLC2A5) (line 36-41).
Is it necessary to italicise the gene names?
Response: According to the Gene/Protein Nomenclature Guidelines and Requirements, for Human, non-human primates and domestic species, gene symbols need to be italicized and proteins do not. In addition, we looked at published articles in animals and found that the genes were in italics.
Keywords
These words act as additional searching terms and should differ to the ones in your title. The use of different words enhances the search criteria.
Response 1: We added and modified maternal nutrition and small intestinal development as the keywords according to your suggestion (line 47-48).
Introduction.
Th information presented here is good and very interesting; however, I strongly recommend rewording the second paragraph for clarification of your arguments. The English expression needs some attention.
Response: We amended the second paragraph to clarify our arguments according to your suggestion (line 79-84).
Methods
Table 1: the animal number is confusing. 16 twin-bearing ewes, why do you have 15 twin lambs? Why are there not 32 lambs? Same for the triplets. If there 5 triplet-bearing ewes, why were there only 6 triplet lambs? Surely you’d want to pair the lamb data to the ewe.
Response: In this experiment, 66 ewes gave birth to twin or triple lambs, and then we selected a total of 65 lambs which birth weight close to the average birth weight of each group for slaughter. For example, a total of 16 ewes in the control group gave birth to 32 twin-born lambs. Then, 15 newborn twin lambs in the control group were slaughtered to study the effect of folic acid on the small intestine development, and the remaining lambs were kept for further study. The same for the other groups.
There is no mention here of the animals actual daily feed intake? Did all the animals consume their daily mixed ration?
Response: DMI was similar throughout the entire gestation period in all treatment groups. We added this information in line 119-120. All ewes were fed total mixed ration every day.
As FA was fed throughout gestation based on ewe body weight, how frequently were the ewes weighed throughout your trial? How often did you need to adjust the concentration FA to account for LW changes, and how did you ensure the ewes were receiving the accurate FA concentration throughout gestation?
Response: In the study, folic acid was supplemented into maternal diet according to feed intake of ewes. We adjusted the amount of TMR and FA supplementation each day according to the previous day's diet surplus. This ensures that FA levels were taken in the intended supplemental level.
Given that the triplet-bearing ewes were significantly lighter postpartum, did you monitor the ewe’s body condition score throughout gestation?
Response: All ewes gained weight postpartum compared to their initial weight. The average initial weight of ewes gave birth to twin and triplet lambs were 44.00±0.53 kg and 44.20±0.57 kg, respectively. The average postpartum weight of twins and triplets were 55.19±0.75 and 51.92±1.01 kg, respectively. In addition, we did not measure the weight and body condition score of ewes in order to minimize the stimulation to pregnant ewes.
Again, the English expression needs some attention. For example. Line 104: Ewes were free access to water? Suggest reword
Response: We replaced “Ewes were free access to water” by “And clean water was available for the animals all the time” according to your suggestion in line 105.
The statistical analysis is too simple for the results presented. Please provide explicit details of the statistical analysis, including the variables measured.
Response: In line 169-177, we changed statistical analysis to “The data of weight and morphology of the small intestine and genes expression were analyzed by the generalized linear model (GLM)of statistical package SPSS version 23.0 (SPSS, IBM, Inc., Chicago, IL, USA) to assess the effects of litter size (TW and TR) and dietary FA supplementation levels (CON, F16 and F32). Treatment means were compared by Duncan’s new multiple range test using a significance level of P < 0.05. Polynomial analysis was conducted to test the linear or quadratic response to the levels of FA supplementation in the diet at 0.05 level. The variables in different litter size was compared by independent-samples T test at 0.05 level. Results are shown as mean ± SEM. The relationships between genes and phenotypes for the various treatments were examined using correlation analysis.”.
Results.
If you intend to use the term tendency you should include the F-statistics information to support your claim that these results were in fact tended towards significance. In saying that, a quick change of your significance value from P < 0.050 to P < 0.05 in the statistical analysis section would remove the need to use the term tendency and you could state there was a significant difference.
Response: Thanks for your suggestion and we deleted this sentence about term tendency.
Line 170: triplet -born. Remove the space before the hyphen.
Response: We removed the space before the hyphen according to your suggestion in line 189.
Line 172: improved is a subjective term. Use increased, was higher, thicker to quantify your results and leave the subjective terms for the discussion.
Response: According to your suggestion, we replaced “With the increase of FA level, the thickness of muscle layer improved linearly (P = 0.019)” with “The jejunal muscle layer thickness was increased linearly with the increase of FA level (P = 0.019).” line 190-191.
Figure 1: lacks information about the specific morphological characteristics. Morphological characteristics is vague. Can you be a bit more specific.
Response: We added the schematic diagram of small intestinal morphological parameters as Figure 1.
The significant correlations were moderate at best and should be stated as such.
Response: I'm sorry that I can't understand what’s your meaning, but we replaced “We also determined the correlation between genes expression and phenotypes (Table 9, Table 10). In duodenum, the expression of MGAM (P = 0.007), LCT (P = 0.002), SGLT1 (P = 0.001), SI (P = 0.004), SLC2A5 (P = 0.034) were positively correlated with crypt depth” by “We also determined the correlations between gene expression and phenotype parameters (villus height, thickness of muscle layer and crypt depth) of the duodenal and jejunal tissues (Table 5 and Table 6). In duodenum, the expression of MGAM, LCT, SGLT1, SI, SLC2A5 were significantly positively correlated with crypt depth (P < 0.05)” in line 258-261.
You mentioned you conducted a regression analysis yet you have not presented those results. Was there any bias in your data?
Response: There is only Pearson correlation analysis in the data analysis, no regression analysis. I am sorry for my vague description caused your misunderstanding.
Discussion
The discussion is well written with grammatical minor errors.
244: in in. Remove the repeated word
Response: We removed the repeated word according to your suggestion.
259: switching from FA to folic acid, then back to FA. Be consistent.
Response: We replaced “folic acid” by “FA” according to your suggestion in line 296.
Reference
Please cross-check your citation information correctly. There are some author details which have been presented incorrectly.
Response: We collated and modified all the references according to your suggestion.

Reviewer 2 Report
Could you present economic information on the treatment?
In what production systems could apply this finding? What producer's profile?
What studies do you recommend to continue this research?
Author Response
Dear Reviewer,
Thank you for helping us improve the quality of manuscript (ID: animals-963445). We gratefully appreciate the constructive comments made by you. We revised the manuscript thoroughly according to your comments. Any changes of the manuscript were indicated in red in the revised manuscript and listed as follows.
Reviewer 2
Comments and Suggestions for Authors
Could you present economic information on the treatment?
Response 1: The price of folic acid is 7690 CNY /kg. The dry matter intake of each sheep during pregnancy is about 132kg. The folic acid feeding amount of each sheep during gestation is 2.1g (132 kg ×16 mg/kg÷1000=2.1g). The folic acid cost of each ewe during gestation is 16.15 CNY in F16 group, and the daily cost of each ewe in F16 group is 0.11 CNY. The cost of folic acid during gestation was 32.3 CNY per ewe in F32 group, and the daily cost of each ewe in F32 group is 0.22 CNY, using the same method.
In what production systems could apply this finding? What producer's profile?
Response 2: This finding can be used on any sheep farm which breed with large litter size, and we have applied for two patents for the series results of this study. According to another of our study (Wang et al., 2019), maternal folic acid supplementation could improve the birth weight of triplets. That is, it can indirectly improve the survival rate of lambs. So, the bigger the sheep farm, the bigger the profit. In addition, large-scale farms are better equipped to perform synchronous estrus and uniformly manage pregnant ewe so that folic acid can be added to the diet.
What studies do you recommend to continue this research?
Response 3: Firstly, the effect of folic acid on intestinal development should be better verified by in vitro cell culture test. And then, further studies on the mechanism of folic acid regulation of intestinal development were carried out by gene modification assay.
References
Wang, B.; Li, H.Q.; Li, Z.; Jian, L.Y.; Gao, Y.F.; Qu, Y.H.; Liu, C.; Xu, C.C.; Li, Y.X.; Diao, Z.C.; Lu, W.; Yu, Y.; Machaty, Z.; Luo, H.L. Maternal folic acid supplementation modulates the growth performance, muscle development and immunity of Hu sheep offspring of different litter size. J. Nutr. Biochem. 2019, 70, 194-201.
Reviewer 3 Report
In this study the authors investigated the effect of maternal folic acid supplementation on lambs’ small intestinal development, where they measured different parameters of the small intestines as gene expression. The authors present a good experimental design and also obtained interesting results. However, as I have highlighted in the comments, in several times the authors only picked on a few of the results without considering the whole picture of their results. I would suggest the authors revise their manuscript, by presenting some their results as bar or line plots instead of presenting everything as tables. This would allow them to visualize the trends in their results that they are missing in the results but as well as the discussion. Additionally, the discussion needs to be greatly improved and be based on the authors data(which they have) rather than just reviewing other people’s, not saying they should not review other studies, however they can use them to support where they do not have hard data/results to so or as a supplement. The conclusions need to be improved to be stronger and cover the study.
Simple summary
Line4: Replace “……lambs of different litter size” with “lambs from differing litter size”.
Line16-18: what do you mean by this “newborn lambs supplemented with FA during maternal gestation.” Please rephrase or rewrite, in your experiment it’s the mothers that were supplemented and not the lambs.
Line18-20: What do you mean by “The supplementation of FA could optimize…..” are you trying to report results by this sentence or giving implication of your results? If its implication, then where are your results in this summary. Please rephrase without using “could” if these are results.
Abstract
Line24: Add “maternal or ewe” between “of” and “dietary”.
Line29-30: Change “(16 or 32 mg FA per kg DM, F16 29 and F32)” to (16 or 32 mg FA per kg DM, i.e. F16 29 and F32)”
Line33: Add “was” between “offspring” and “enhanced”.
Line34: Remove “genes”, should read as follows “Meanwhile, the expression of IGF-I…………..”.
Line36-37: Delete this line “However, litter size 36 had different effects on intestinal phenotypes and genes expression.” You are not making comparison between litter size and FA supplementation so this sentence is redundant.
Line38-39: “and the expression of IGF-I (P < 0.05), SI (P < 0.05) and SLC2A5 (P < 0.01) were lower than triplets.” What does the P-values in the brackets imply, significant differences? If so, then write it, otherwise they do not seem to mean a lot in the current state.
Line42-44: Delete this “and the expression of IGF-I (P < 0.05), SI (P < 38 0.05) and SLC2A5 (P < 0.01) were lower than triplets.” This is not new information you already said it before.
Introduction
Line50-52: Reference is missing for this information “Folic acid is one of 50 the water-soluble B vitamins and performs critical functions in amino acid metabolism, DNA 51 synthesis and repair, as well as gene expression and modification.”
Line59-61: Add “the” between “To” and “fetus”.
Line59-61: What is the reference of this information “To fetus, the intestinal development is very sensitive to the intrauterine environmental variation during whole gestation period, and intestine undergoes dynamic changes throughout the life of the animal.”?
Line64: change “leaded” to “led”.
Line66: Change “synthetic” to “synthesize”
Line69: Change “satisfy” to “satisfactorily”.
Line71: Any citation about prolificacy of the Hu sheep?
Line74-76: Please rewrite this without using “Then” instead use “Additionally” or another word.
Line88-90: This is not clear, please state exactly how many animals received which diet.
Materials and Methods
Line91: Please state how many animals were removed from the study/analysis since they were non-pregnant.
Line92: How many animals produced twins and how many had triplets.
Line95-97: What do the numbers in brackets mean in Table 1 (e.g. TW-CON (16, 15)? Please define them in the legend below the table. If these are animals of the same grouping them why two numbers and not one number?
Line103: Add “all” before “animals”.
Line106-108: Please define the units of the figures in table 2, are they percentages or something else? You could have two tables for this table to avoid confusion.
Line121: When were the animals euthanized? I mean, how many days after lambing? Probably the animals were not born on the same day, meaning some were days older than the other, hence I would like to see the effect in the sampling age on the studied phenotype.
Line128: what do you mean by “for following measurements”. Please rephrase.
Line141: What the quality of your extracted RNA and the average concentration you must have gotten that from the nanodrop.
Line142: Change “First chain” to “First strand”
Line142-144: Please state exactly what you did, not just stating the kit. State the quantity of the materials you used in the qPCR experiment like the amount of RNA used for reverse transcription. Write this part stating step by step of what you did, not just writing the kit.
Line148-150: How did you make the primers? Were they custom made primers that you bought, please state that, and name the company? Provide the details of the primers like the primer melting temperature, expected fragment size etc in Table 3.
Line146-147: Provide the citations reporting these genes to be involved in the biological functions you are mentioning.
Line155: How do you analyse “All results” It should be “All data”.
Line155: write GLM in full (general linear model) before using the short form.
Line157: How did you perform the polynomial analysis?
Line158: Replace “were analyzed” with “were performed”.
Results
Line162: Replace “was” with “is”.
Line164-165: How did you test the linearity when FA was given the fact that the you only gave the FA supplementation in just three categorical groups.
Line166: These should be results not data, please rephrase and use the correct word.
Line166-167: “respectively” is missing at the end of this sentence.
Table 4 : Break the lines below the treatment tittles, to separate the treatment titles (i.e. break the line between Twins and Triplets, break the line between litter size and diet.)
Line167-168: Rephrase this “The duodenal muscle layer thickness showed a significant interaction effect between litter size and FA supplementation (P = 0.011)” to “There was a significant (P = 0.011) effect of the interaction between litter size and FA supplementation on duodenal muscle layer thickness of the lambs.
Line168: What is “it”.
Line168-170: Please rewrite this, I cant understand what you are trying to communicate.
Line171: Delete “but no significant interaction.”
Line172:” thickness of muscle layer” Thickness of which muscle?
Line171-172: Please rewrite, starting with “The thickness…..”
In your results please report about the differences between F16 and F32 groups. you can use bar plots to present some of these comparisons between CON, F16 and F32. Tell the readers whether you think it is better to supplement with 16 or 32 mg of FA or it doesn’t matter, i.e. no difference between these supplementation levels.
Line193: This statement should be changed, it not true, mention the specific genes that were affected, according to your results in Table 6.
Line193-194: Rewrite this, Use “The interaction between FA and Litter size had significant effect..” Do not use “prominent” it doesn’t have the meaning as significant here.
Line194-196: To avoid confusion please in all your reporting of the results, report the main effects/treatments first before reporting the interaction.
Line194-195: What do you mean by this “The duodenal expression of IGF-I was not affected by FA supplementation in twin-born lambs (P = 0.310)” I do not find this in your table 6.
Line193-199: Please highlight the results that are most interesting, do not try to rewrite everything from the tables. Additionally, some of the P-values you are referring to I can not find them in the table 6. Please rewrite this section.
Line200: Replace “were” with “are”
Line200-201: Rephrase this sentence “In the duodenum, the remarkable linearly enhancement of the expression of BCL-200 2 was affected by FA supplementation (P = 0.027)” it is grammatically wrong.
Line201: “In the jejunal” rephrase this as “For the jejunum……”
Line201-202: Like in the previous section you are reporting the interaction effect without even mentioning the effect of the main studied treatments. Please revise this.
Line202: “It showed a quadratic response to FA supplementation in twin lambs (P = 0.034).” What is “it” in this sentence? Every sentence should be a stand-alone information, if you want to make it related to the preceding sentence, then use joining words. This is confusing please rephrase to make this sentence clear.
Line203: I see you are trying to use different words and formats of reporting your results, however its not helping, keep it simple and it is okay to use same simple words like significant, significantly, substantial effect in all your results where necessary.
Line203: Please rephrase this “The results in Table 8 revealed the effect dietary FA supplementation during gestation on small intestinal digestion-related genes expression of newborn lambs.” Some results actually do not show any effect, so I do not understand what you are trying to mean.
Line204-205: Firstly, a similar trend of writing results is again repeated here, writing the effect of the interaction before or even without reporting the effect of the main effects, even if it is not significant or one of them is not significant report it them move on to report the interaction effect. You do not have to report each P-value in the bracket besides the gene, since P<0.05 was the threshold, you can obviously write that “the interaction between litter size and FA supplementation had significant (P < 0.05) effect on SGLT1 in the duodenal tissue, and LCT and SLC2A5 in the jejunal tissue. Please rewrite this.
Line204: Even in other parts of the manuscript where you have used duodenal and jejunal, please write “tissue” after “duodenal” or “jejunal”.
Line206 “Meanwhile, the addition of FA had no effect on them….” Who or what is “them”? as noted above, each sentence should be stand-alone. Please rephrase and state the genes names instead of “them”.
Line208: What do you mean by this “Moreover, the effect of litter size on duodenal gene expression was also prominent.” Is this for all the genes you profiled in this tissue? I believe no, if I am right please be specific and mention the gene you are referring to in this sentence. “Prominent” is not the right word to use here, use significant where necessary.
Line208-209: No need for putting P values in brackets besides the genes, just write as you did in the sentence Line207-208.
Line228-229: Rephrase this to, “We also determined the correlations between gene expression and the measured phenotypes (villus height, thickness of muscle layer and crypt depth) of the duodenal and jejunal tissues (Table 9 and Table 10).
Line229-230: Rephrase this to “In duodenum, the expression of MGAM , LCT, SGLT1, SI, SLC2A5 were significantly (P<0.05)positively correlated with crypt depth.
Do not concentrate on picking up only significant values, internalize your results and give a general overview of your results (or the trends in the results) than just picking up a few significant ones and ignore the rest. For example if you look at table 9, you can see that though the correlations between thickness of muscle layer and gene expression in the duodenum tissue is not significant, you can see a general trend of negative correlation, Please report this. Also villus height seems to be positively correlated with gene expression in the duodenum. Please consider similar observations in the jejunum and report the general trends and these make it easy for you to even compare the two tissues in terms of correlations.
In your results please report about the differences between F16 and F32 groups. you can use bar plots to present some of these comparisons between CON, F16 and F32. Tell the readers whether you think it is better to supplement with 16 or 32 mg of FA or it doesn’t matter, i.e. no difference between these supplementation levels. Add some of these figures in the manuscript and you make the tables supplementary.
For all the tables you presented please provide standard error of each mean in the table.
Please add a line separating gene profiles of the two tissues you studies, tables seem a bit difficult to follow.
Discussion
Line242-243: Provide a citation for this “As a vital methyl-donor, FA plays a role in the maintenance and development of the fetus during gestation.”
Line243-244:Please rephrase this sentence to something like “Maternal supplementation of methyl-donors such as FA to has been reported to decrease incidence of pregnancy failure and increase litter size in early parity sows [15].”
Line244-246: Please provide the citation for this.
Line249: Replace “discovered” with “reported”.
Line254-256: Please, try to link this sentence to what you have written before it.
Line242-256: Is a very good piece review paragraph but needs to be linked to your study results, not just review. Please try to add the link.
Line271-274: Therefore according to your results would it be better to supplement triplet carrying mothers with a higher FA level, such as 32mg that you used in your study, does it make a difference supplementing 16mg or 32mg to twins or triplets. Please say something and discuss about this here, you the results and they seem to show some interesting differences. Try plotting the phenotype mean values of your results and internalize this.
Line275-277: If this statement is true, then how else do you think FA affects morphological differences other than through influencing gene expression. I think this is general knowledge that for a molecule to cause phenotypic difference, it has to affect gene expression that results into the observable phenotypic differences. Rephrase please.
Line282-283: I barely understand what you are trying to communicate by this sentence. You should highlight the results before you discuss them, instead of discussing or reviewing several citation, and then bring in results. The link between this sentence and the preceding sentences is barely recognizable.
Line286-287: What do you mean by this “the fetal programming determined by epigenetic 286 modification can be adjusted by influencing the mRNA abundance” What modifies the mRNA abundance, without changing the epigenome. I do not think mRNA abundance can be influenced by not affecting epigenome first. Please this.
Line289-291: This explanation does not really sound fitting here, try to internalize your data before finding explanation for a few results you pick on, look at the large picture of your results. For example in your study you three levels of FA supplementation with one being zero supplementation, I would you to highlight what trends do you see when you supplement with 32mg than 16mg. Do mean that supplement 32mg also still allows for FA deficiency? Please spend some time reinternalizing your results as a whole not just bits. Could you discuss the observed differences between the studied two tissues, duodenum and the jejunum if any?
Line292: Which previous results are you referring to here? Any citations?
Line293-294: Please write based on your results, this statement “the intestinal morphology of triplets was poorer than that of twins, resulting in high expression of development-related genes.” Is not true according to all the results you present in this paper. Please internalize your results and states where increases where observed and where decreases were observed and then discuss that.
Line293-294: “the intestinal morphology of triplets was poorer than that of twins, resulting in high expression of development-related genes.” How can gene expression result from a phenotype? It is the other way round please think about a better discussion after internalizing your results.
Line295-296: “Gene expression can be reversed, but phenotypic reversal is very difficult.” Then? This sentence sounds redundant.
Line308-310: Please delete this “For example, apoptotic cells in the small intestine were significantly increased in celiac disease, suggesting that increased apoptotic cells were associated with villi atrophy in the disease [33]”
Line315-318: Provide a citation/citations for this information.
Line325-327:This statement is not true for all the cases, in some cases controls actually had higher expression of LCT than FA supplemented (see Triplets CO vs F32 duodenum) or (see Twins CON vs F16). For SGLT1 (see Triplets CO vis F16 or F32) and (Triplet CO vs F32).You could say that, “In this study, dietary FA supplementation in gestation generally increased expression of …………” Please review your results and discuss them as a whole, not picking a few.
Line327-329: “which is consistent with what we found in terms of small intestine development-related gene expression.” What do you mean by this? Please rephrase this doesn’t add any new information.
Line332-335: I still do not get the logic of your explanation, here you are talking in terms of absorption genes. Try to link your results together to get good explanation for your results. For example in this case an increased expression of absorption related genes in triplets could actually be compensation for the small size (in terms of weight look at Table 4) absorptive surface area, since in your results you mention that triplet had smaller small intestines as compared to twins. In this case the animal would compensate for this reduced size. I again suggest you revisit your results and consider discussing them as a whole, you have very good results from which you can draw good discussion.
Conclusions
Line343-349: Please revise the conclusions giving general conclusions on the treatments you studied and what are the implications of your observations. You categorized the genes you studied, please general conclusion on those groups even if its just a trend, not just picking on just a few.
Author Response
Response to Reviewer 3 Comments
Dear Reviewer,
Thank you for helping us improve the quality of manuscript (ID: animals-963445). We gratefully appreciate the constructive comments made by you. We revised the manuscript thoroughly according to your comments. Any changes of the manuscript were indicated in red in the revised manuscript and listed as follows.
Reviewer 3.
Comments and Suggestions for Authors
In this study the authors investigated the effect of maternal folic acid supplementation on lambs’ small intestinal development, where they measured different parameters of the small intestines as gene expression. The authors present a good experimental design and also obtained interesting results. However, as I have highlighted in the comments, in several times the authors only picked on a few of the results without considering the whole picture of their results. I would suggest the authors revise their manuscript, by presenting some their results as bar or line plots instead of presenting everything as tables. This would allow them to visualize the trends in their results that they are missing in the results but as well as the discussion. Additionally, the discussion needs to be greatly improved and be based on the authors data(which they have) rather than just reviewing other people’s, not saying they should not review other studies, however they can use them to support where they do not have hard data/results to so or as a supplement. The conclusions need to be improved to be stronger and cover the study.
Simple summary
Line4: Replace “……lambs of different litter size” with “lambs from differing litter size”.
Response: We replaced “lambs of different litter size” by “lambs from differing litter sizes” according to your suggestion (line 4-5).
Line16-18: what do you mean by this “newborn lambs supplemented with FA during maternal gestation.” Please rephrase or rewrite, in your experiment it’s the mothers that were supplemented and not the lambs.
Response: We replaced “lambs supplemented with FA during maternal gestation” by “lambs whose mothers supplemented FA during gestation” according to your suggestion in line 19.
Line18-20: What do you mean by “The supplementation of FA could optimize…..” are you trying to report results by this sentence or giving implication of your results? If its implication, then where are your results in this summary. Please rephrase without using “could” if these are results.
Response: We replaced “could optimize” by “improved” according to your suggestion in line 20.
Abstract
Line24: Add “maternal or ewe” between “of” and “dietary”.
Response: Thanks for your suggestion and we added “maternal” between “of” and “dietary” in line 25.
Line29-30: Change “(16 or 32 mg FA per kg DM, F16 29 and F32)” to (16 or 32 mg FA per kg DM, i.e. F16 29 and F32)”
Response: According to your suggestion, we change “(16 or 32 mg FA per kg DM, F16 and F32)” to “(16 or 32 mg FA per kg DM, i.e. F16 and F32)” in line 31.
Line33: Add “was” between “offspring” and “enhanced”.
Response: We added “was” between “offspring” and “enhanced” according to your suggestion (line 35).
Line34: Remove “genes”, should read as follows “Meanwhile, the expression of IGF-I…………..”.
Response: We remove “genes” between “the” and “expression” (line 36).
Line36-37: Delete this line “However, litter size 36 had different effects on intestinal phenotypes and genes expression.” You are not making comparison between litter size and FA supplementation so this sentence is redundant.
Response: We deleted this sentence and added “Moreover,” before “The small intestinal” in line 39.
Line38-39: “and the expression of IGF-I (P < 0.05), SI (P < 0.05) and SLC2A5 (P < 0.01) were lower than triplets.” What does the P-values in the brackets imply, significant differences? If so, then write it, otherwise they do not seem to mean a lot in the current state.
Response: We added “significantly” between “were” and “lower” (line 41).
Line42-44: Delete this “and the expression of IGF-I (P < 0.05), SI (P < 38 0.05) and SLC2A5 (P < 0.01) were lower than triplets.” This is not new information you already said it before.
Response: Thanks a lot for your comments. Before this sentence we compared the effects of folic acid supplementation on the expression of IGF-I, BCL-2, BAX and SGLT1 in line 36-38, and this sentence we compared the effects of litter size on the expression of IGF-I, SI and SLC2A5 in line 40-41.
Introduction
Line50-52: Reference is missing for this information “Folic acid is one of 50 the water-soluble B vitamins and performs critical functions in amino acid metabolism, DNA 51 synthesis and repair, as well as gene expression and modification.”
Response: we added a reference is missing for this information (line 55).
Line59-61: Add “the” between “To” and “fetus”.
Response: We deleted this sentence and added “The ruminant offspring intestinal tissues are responsive to maternal nutrition during gestation [8]” in line 62-63.
Line59-61: What is the reference of this information “To fetus, the intestinal development is very sensitive to the intrauterine environmental variation during whole gestation period, and intestine undergoes dynamic changes throughout the life of the animal.”?
Response 13: We replaced “To fetus, the intestinal development is very sensitive to the intrauterine environmental variation during whole gestation period, and intestine undergoes dynamic changes throughout the life of the animal.” by “The ruminant offspring intestinal tissues are responsive to maternal nutrition during gestation.” and added a reference of this sentence in line 62-63.
Line64: change “leaded” to “led”.
Response 14: We changed “leaded” to “led” according to your suggestion in line 66.
Line66: Change “synthetic” to “synthesize”
Response 15: We changed “synthetic” to “synthesize” according to your suggestion (line 68).
Line69: Change “satisfy” to “satisfactorily”.
Response 16: According to your suggestion, we replaced “satisfy” by “satisfactorily” (line 71).
Line71: Any citation about prolificacy of the Hu sheep?
Response: We added a reference of this sentence according to your suggestion in line 75.
Line74-76: Please rewrite this without using “Then” instead use “Additionally” or another word.
Response: We changed “Then” to “Additionally,” according to your suggestion (line 77).
Line88-90: This is not clear, please state exactly how many animals received which diet.
Response: We changed “and randomly divided into three groups after artificial insemination.” to “After artificial insemination, they were randomly divided into three groups with forty ewes in each group”, and replaced “All animals were fed one of three diets in individual enclosures (size: 1.5×3 m2): control (CON), 16 (F16) or 32 (F32) mg/kg (DM basis) of rumen-protected FA into the control diet until parturition.” by “The three treatments were supplemented with 0 (CON), 16 (F16) or 32 (F32) mg/kg (DM basis) of rumen-protected FA into the control diet until parturition. All the animals were raised in individual pens (size: 1.5×3 m2).” according to your suggestion (line 92-94).
Materials and Methods
Line91: Please state how many animals were removed from the study/analysis since they were non-pregnant.
Response: Thirty-five ewes without pregnancy were eliminated throughout the trial period, and we added this information to the manuscript and we added this information in line 96.
Line92: How many animals produced twins and how many had triplets.
Response: The number of ewes produced twins and triplets were 39 and 27, respectively (line 117-118).
Line95-97: What do the numbers in brackets mean in Table 1 (e.g. TW-CON (16, 15)? Please define them in the legend below the table. If these are animals of the same grouping them why two numbers and not one number?
Response: We changed Table 1 to “The number of ewes gave birth to twin and triplet were 16 and 5; 13 and 13; 10 and 9 for the CON, F16 and F32 groups, respectively. The number of twins and triplets used in this experiment were 15 and 6; 11 and 11; 8 and 14 for the CON, F16 and F32 groups, respectively” to io make it clear in line 117-119.
Line103: Add “all” before “animals”.
Response: We added “all” before “animals” (line 104).
Line106-108: Please define the units of the figures in table 2, are they percentages or something else? You could have two tables for this table to avoid confusion.
Response: The unit of the figures is percentage, which is indicated at the end of the table caption in line 107.
Line121: When were the animals euthanized? I mean, how many days after lambing? Probably the animals were not born on the same day, meaning some were days older than the other, hence I would like to see the effect in the sampling age on the studied phenotype.
Response: These lambs were measured for weight, body size and other phenotypes immediately after birth and then euthanized immediately. That is, the animals were not slaughtered on the same day.
Line128: what do you mean by “for following measurements”. Please rephrase.
Response: We changed “following measurements” to “qRT-PCR test,” according to your suggestion (line 132).
Line141: What the quality of your extracted RNA and the average concentration you must have gotten that from the nanodrop.
Response: The values of OD260/280 ratio were all between 1.8 - 2.0, which could meet the requirement of RNA purity. The concentration range of RNA is 158.2 - 450.3 ng/μl, so it is possible to establish a reaction system of 20 μl containing 50 ng-2 μg RNA in reverse transcription test.
Line142: Change “First chain” to “First strand”
Response: We changed “First chain” to “First strand” according to your suggestion (line 146).
Line142-144: Please state exactly what you did, not just stating the kit. State the quantity of the materials you used in the qPCR experiment like the amount of RNA used for reverse transcription. Write this part stating step by step of what you did, not just writing the kit.
Response: We added “The 25 μL Real-Time PCR reaction mix contained 12.5 μL 2x SuperReal Premix Plus, 0.5 μL Forward Primer (10 μM), 0.5 μL, Reverse Primer (10 μM), 1 μL cDNA and 10.5 μL RNase-free ddH2O” between “FastQuant DNA first-chain synthesis kit (Tiangen, Biochemical Technology Co., Ltd, Beijing, China)” and “The qRT-PCR was performed on an iQ5 system”, and added “ The 25 μL Real-Time PCR reaction mix contained 12.5 μL 2x SuperReal Premix Plus, 0.5 μL Forward Primer (10 μM), 0.5 μL, Reverse Primer (10 μM), 1 μL cDNA and 10.5 μL RNase-free ddH2O” between “duodenal and jejunal segment of newborn lambs.” and “The sequences of the forward” to explain the specific reaction system (line 147-157).
Line148-150: How did you make the primers? Were they custom made primers that you bought, please state that, and name the company? Provide the details of the primers like the primer melting temperature, expected fragment size etc in Table 3.
Response: The design and synthesis of primers was handled by Shanghai Generay Biotech Co., Ltd, Shanghai, China (line 156-157). We added the primer melting temperature and product size in Table 2.
Line146-147: Provide the citations reporting these genes to be involved in the biological functions you are mentioning.
Response: We added the citations reporting these genes about their biological functions (line 154).
Line155: How do you analyse “All results” It should be “All data”.
Response: We replaced “results” by “data” according to your suggestion in line 169.
Line155: write GLM in full (general linear model) before using the short form.
Response: We added general linear model as the full name of GLM in line 170.
Line157: How did you perform the polynomial analysis?
Response: We used one-way ANOVA, clicking contrasts button, and selected polynomial, and then selected degree: quadratic. We used the LSD test at a 0.05 significance level.
Line158: Replace “were analyzed” with “were performed”.
Response: We changed this sentence to “The relationships between genes and phenotypes for the various treatments were performed using correlation analysis.” (line 175-177).
Results
Line162: Replace “was” with “is”.
Response: We replaced “was” by “is” according to your suggestion in line 180.
Line164-165: How did you test the linearity when FA was given the fact that the you only gave the FA supplementation in just three categorical groups.
Response: We test the linearity of FA supplementation when we did polynomial analysis.
Line166: These should be results not data, please rephrase and use the correct word.
Response: According to your opinion, we changed “data” to “results” in line 184.
Line166-167: “respectively” is missing at the end of this sentence.
Response: We added “, respectively” at the end of this sentence (line 185).
Table 4: Break the lines below the treatment tittles, to separate the treatment titles (i.e. break the line between Twins and Triplets, break the line between litter size and diet.)
Response: We broken the lines below the treatment tittles, to separate the treatment titles according to your opinion.
Line167-168: Rephrase this “The duodenal muscle layer thickness showed a significant interaction effect between litter size and FA supplementation (P = 0.011)” to “There was a significant (P = 0.011) effect of the interaction between litter size and FA supplementation on duodenal muscle layer thickness of the lambs.
Response: We replaced “The duodenal muscle layer thickness showed a significant interaction effect between litter size and FA supplementation (P = 0.011)” by “There was a significant (P = 0.011) effect of the interaction between litter size and FA supplementation on duodenal muscle layer thickness of the lambs” according to your suggestion in line 185-186.
Line168: What is “it”.
Response: We changed “It” to “The thickness of duodenal muscle layer” to make the sentence easy to understand (line 186-187).
Line168-170: Please rewrite this, I cant understand what you are trying to communicate.
Response: According to your opinion, we changed “It increased linearly……triplet -born lambs (P = 0.134)” to “The thickness of duodenal muscle layer increased linearly with maternal dietary FA supplementation in the twin-born lambs (P < 0.001), while FA supplementation had no influence on the duodenal muscle layer thickness among the triplet -born lambs (P = 0.134).” in line 186-189.
Line171: Delete “but no significant interaction.”
Response: We deleted “, but no significant interaction” according to your suggestion.
Line172:” thickness of muscle layer” Thickness of which muscle?
Response: We changed “thickness of muscle layer” to “jejunal muscle layer thickness” according to your suggestion (line 190).
Line171-172: Please rewrite, starting with “The thickness…..”
Response: We replaced “With the increase of FA level, the thickness of muscle layer improved linearly (P = 0.019).” by “The jejunal muscle layer thickness was increased linearly with the increase of FA level (P = 0.019).” according to your suggestion in line 190-191.
In your results please report about the differences between F16 and F32 groups. you can use bar plots to present some of these comparisons between CON, F16 and F32. Tell the readers whether you think it is better to supplement with 16 or 32 mg of FA or it doesn’t matter, i.e. no difference between these supplementation levels.
Response: In order to make the results more intuitive, we changed Table 5, Table 7 and Table 8 into bar charts according to your suggestion. We also compared F16 and F32, but it is not difficult to see that there is no uniform change rule between each indicator of F16 and F32, so we cannot conclude which addition level has a better effect on the development of small intestine in offspring. And we added “Finally, since most of the effects of F16 and F32 on small intestine morphological development were not significantly different, it was not possible to determine which dose was better for small intestine development in newborn lambs” (line 311-313) and “As for which F16 or F32 has the better effect, it cannot be judged based on the existing data and deserve future study.” (line 386-387) in the discussion and conclusion.
Line193: This statement should be changed, it not true, mention the specific genes that were affected, according to your results in Table 6.
Response: We replaced “Litter size and dietary FA supplementation during pregnancy influenced the development-related genes expression in the newborn lambs (Table 6)” by “The effect of litter size and FA supplementation on intestinal development-related genes expression is shown in Table 4” in line 226.
Line193-194: Rewrite this, Use “The interaction between FA and Litter size had significant effect.” Do not use “prominent” it doesn’t have the meaning as significant here.
Response: We replaced “The expression of IGF-I showed a prominent interaction effect between litter size and FA supplementation in duodenum (P = 0.005)” with “The interaction between FA supplementation and litter size had significant effect on the expression of IGF-I in duodenum (P = 0.005)” in line 226-227.
Line194-196: To avoid confusion please in all your reporting of the results, report the main effects/treatments first before reporting the interaction.
Response: According to your suggestion, we changed “The duodenal expression of IGF-I was not affected by FA supplementation in twin-born lambs (P = 0.310), but was increased linearly with dietary FA supplementation in triplet-born lambs (P = 0.039)” to “The duodenal expression of IGF-I was increased linearly with dietary FA supplementation in triplet-born lambs (P = 0.039), but was not affected by FA supplementation in twin-born lambs (P = 0.310)” in line 227-228.
Line194-195: What do you mean by this “The duodenal expression of IGF-I was not affected by FA supplementation in twin-born lambs (P = 0.310)” I do not find this in your table 6.
Response: In twins or triplets, P-values with significant difference after FA supplementation were marked in the footnotes of the table, while P-values without significant difference were not marked. So, the effect of FA supplementation in twin-born lambs on the duodenal expression of IGF-I and the P-values cannot be found in Table 4.
Line193-199: Please highlight the results that are most interesting, do not try to rewrite everything from the tables. Additionally, some of the P-values you are referring to I can not find them in the table 6. Please rewrite this section.
Response: To stress the key point, we deleted “in the FA supplementation groups, the duodenal expression of IGF-I in lambs with litter size of triplet was significant improved than twins (P < 0.001). Similarly,”.
Line200: Replace “were” with “are”
Response: According to your suggestion, we changed “were” to “are” (line 232).
Line200-201: Rephrase this sentence “In the duodenum, the remarkable linearly enhancement of the expression of BCL-200 2 was affected by FA supplementation (P = 0.027)” it is grammatically wrong.
Response: We replaced “In the duodenum, the remarkable linearly enhancement of the expression of BCL-2 was affected by FA supplementation (P = 0.027)” with “In the duodenum, maternal diet FA supplementation significantly linearly increased the expression of BCL-2 (P = 0.027)” in line 232-233.
Line201: “In the jejunal” rephrase this as “For the jejunum……”
Response: We changed “In the jejunal” to “For the jejunum” according to your suggestion (line 233).
Line201-202: Like in the previous section you are reporting the interaction effect without even mentioning the effect of the main studied treatments. Please revise this.
Response: We changed “In the jejunal, the expression of BAX of newborn lambs were influenced by the interaction effect between litter size and FA supplementation (P = 0.046). It showed a quadratic response to FA supplementation in twin lambs (P = 0.034).” to “For the jejunum, the expression of BAX of newborn lambs were influenced by the interaction effect between litter size and FA supplementation (P = 0.046), which showed a quadratic response to FA supplementation in twin lambs (P = 0.034)” according to your suggestion in line 233-234.
Line202: “It showed a quadratic response to FA supplementation in twin lambs (P = 0.034).” What is “it” in this sentence? Every sentence should be a stand-alone information, if you want to make it related to the preceding sentence, then use joining words. This is confusing please rephrase to make this sentence clear.
Response: Thanks for your correction, and we changed “. It” to “, which” in line 234.
Line203: I see you are trying to use different words and formats of reporting your results, however its not helping, keep it simple and it is okay to use same simple words like significant, significantly, substantial effect in all your results where necessary.
Response: We changed “Moreover, the effect of litter size on duodenal gene expression was also prominent.” to “Moreover, the effect of litter size on the expression of SI and SLC2A5 was also significant in the duodenal tissue” according to your suggestion (line 240-241).
Line203: Please rephrase this “The results in Table 8 revealed the effect dietary FA supplementation during gestation on small intestinal digestion-related genes expression of newborn lambs.” Some results actually do not show any effect, so I do not understand what you are trying to mean.
Response: We replaced “The results in Table 8 revealed the effect dietary FA supplementation during gestation on small intestinal digestion-related genes expression of newborn lambs” with “The Figure 5 shows the results of small intestinal digestion-related genes expression of newborn lambs.” line 235.
Line204-205: Firstly, a similar trend of writing results is again repeated here, writing the effect of the interaction before or even without reporting the effect of the main effects, even if it is not significant or one of them is not significant report it them move on to report the interaction effect. You do not have to report each P-value in the bracket besides the gene, since P<0.05 was the threshold, you can obviously write that “the interaction between litter size and FA supplementation had significant (P < 0.05) effect on SGLT1 in the duodenal tissue, and LCT and SLC2A5 in the jejunal tissue. Please rewrite this.
Response: We changed “The expression of duodenal SGLT1 (P < 0.001) and jejunal LCT (P = 0.015), SLC2A5 (P = 0.045) were influenced by the interaction effect between litter size and FA supplementation” to “The interaction between litter size and FA supplementation had significant (P < 0.05) effect on the expression of SGLT1 in the duodenal tissue, and LCT and SLC2A5 in the jejunal tissue.” according to your suggestion in line 235-236.
Line204: Even in other parts of the manuscript where you have used duodenal and jejunal, please write “tissue” after “duodenal” or “jejunal”.
Response: We added “tissue” after “duodenal” or “jejunal” according to your suggestion in the entire manuscript.
Line206 “Meanwhile, the addition of FA had no effect on them….” Who or what is “them”? as noted above, each sentence should be stand-alone. Please rephrase and state the genes names instead of “them”.
Response: We changed “them” to “the expression of SGLT1 in the duodenal tissue and SLC2A5 in the jejunal tissue” according to your suggestion in line 238-239.
Line208: What do you mean by this “Moreover, the effect of litter size on duodenal gene expression was also prominent.” Is this for all the genes you profiled in this tissue? I believe no, if I am right please be specific and mention the gene you are referring to in this sentence. “Prominent” is not the right word to use here, use significant where necessary.
Response: According to your suggestion, we changed “Moreover, the effect of litter size on duodenal gene expression was also prominent” to “Moreover, the effect of litter size on the expression of SI and SLC2A5 was also significant in the duodenal tissue” (line 240-241).
Line208-209: No need for putting P values in brackets besides the genes, just write as you did in the sentence Line207-208.
Response: We rewrote this sentence as “The expression of SI and SLC2A5 were elevated in the triplets compared to those in the twins (P < 0.05).” according to your suggestion (line 241).
Line228-229: Rephrase this to, “We also determined the correlations between gene expression and the measured phenotypes (villus height, thickness of muscle layer and crypt depth) of the duodenal and jejunal tissues (Table 9 and Table 10).
Response: We changed the sentence to “We also determined the correlations between gene expression and the measured phenotypes (villus height, thickness of muscle layer and crypt depth) of the duodenal and jejunal tissues (Table 5 and Table 6)” in line 258-260.
Line229-230: Rephrase this to “In duodenum, the expression of MGAM, LCT, SGLT1, SI, SLC2A5 were significantly (P<0.05) positively correlated with crypt depth.
Response: According to your suggestion, we rephrase this to “In duodenum, the expression of MGAM, LCT, SGLT1, SI, SLC2A5 were significantly positively correlated with crypt depth (P < 0.05)” in line 260-261.
Do not concentrate on picking up only significant values, internalize your results and give a general overview of your results (or the trends in the results) than just picking up a few significant ones and ignore the rest. For example if you look at table 9, you can see that though the correlations between thickness of muscle layer and gene expression in the duodenum tissue is not significant, you can see a general trend of negative correlation, Please report this. Also villus height seems to be positively correlated with gene expression in the duodenum. Please consider similar observations in the jejunum and report the general trends and these make it easy for you to even compare the two tissues in terms of correlations.
Response: Thanks very much for your suggestion. We added “In general, although no significant correlations between thickness of muscle layer and gene expression in the duodenum tissue, there was a general trend of negative correlation. In addition, villus height seems to be a non-significant positive correlation with gene expression in the duodenum and jejunum tissue” at the end of this paragraph and explained it in the discussion section (line 264-267).
In your results please report about the differences between F16 and F32 groups. you can use bar plots to present some of these comparisons between CON, F16 and F32. Tell the readers whether you think it is better to supplement with 16 or 32 mg of FA or it doesn’t matter, i.e. no difference between these supplementation levels. Add some of these figures in the manuscript and you make the tables supplementary.
Response: Just like the previous answer, in order to make the results more intuitive, we changed Table 5, Table 7 and Table 8 into bar charts according to comments of reviewer. We also compared F16 and F32, but it is not difficult to see that there is no uniform change rule between each indicator of F16 and F32, so we cannot conclude which addition level has a better effect on the development of small intestine in offspring. And we added “Finally, since most of the effects of F16 and F32 on small intestine morphological development were not significantly different, the optimal level of FA supplementation on intestine development deserves further study.”(line 311-313) and “As for which F16 or F32 has the better effect, it cannot be judged based on the existing data and deserve future study.”(line 386-387) in the discussion and conclusion.
For all the tables you presented please provide standard error of each mean in the table.
Response: We added standard error of each mean in the tables according to your suggestion.
Please add a line separating gene profiles of the two tissues you studies, tables seem a bit difficult to follow.
Response: We added a line separating gene profiles of the two tissues in the tables according to your suggestion.
Discussion
Line242-243: Provide a citation for this “As a vital methyl-donor, FA plays a role in the maintenance and development of the fetus during gestation.”
Response: We added a citation for this information in line 278.
Line243-244: Please rephrase this sentence to something like “Maternal supplementation of methyl-donors such as FA to has been reported to decrease incidence of pregnancy failure and increase litter size in early parity sows [15].”
Response: According to your suggestion, we replaced this sentence with “Maternal supplementation of methyl-donors such as FA has been reported to decrease incidence of pregnancy failure and increase litter size in early parity sows [26].” in line 278-280.
Line244-246: Please provide the citation for this.
Response: We added a citation for this information (line 281).
Line249: Replace “discovered” with “reported”.
Response: We changed “discovered” to “reported” according to your suggestion (line 284).
Line254-256: Please, try to link this sentence to what you have written before it.
Response: We changed it to “The weight of small intestine and its proportion of body weight are important indicators of its development. In the present study, it was shown that the increase of the small intestine/live body weight ratio in FA-supplemented groups was the most direct indicator of improved intestinal development because of maternal dietary FA supplementation” to link the two sentences in line 389-392.
Line242-256: Is a very good piece review paragraph but needs to be linked to your study results, not just review. Please try to add the link.
Response: We have modified parts of this paragraph as shown in the responses above (line 277-392).
Line271-274: Therefore according to your results would it be better to supplement triplet carrying mothers with a higher FA level, such as 32mg that you used in your study, does it make a difference supplementing 16mg or 32mg to twins or triplets. Please say something and discuss about this here, you the results and they seem to show some interesting differences. Try plotting the phenotype mean values of your results and internalize this.
Response: In the results of small intestine morphology, since the interaction between FA and litter size was almost insignificant, we discussed the effects of FA or litter size on small intestine morphology, without considering the difference supplementing 16mg or 32mg to twins or triplets. According to your suggestion, we re-compared the effects of 16 mg and 32 mg on small intestine, and we believed that these two levels had no significant differences in small intestine weight and morphological development, which could not be said that 32 mg had better effects. And we added “Animals with nutritional limitations in the fetus tend to have poorer morphological development in the small intestine after birth than normal animals” and “Finally, since most of the effects of F16 and F32 on small intestine morphological development were not significantly different, it was not possible to determine which dose was better for small intestine development in newborn lambs” in this paragraph (line 305-313).
Line275-277: If this statement is true, then how else do you think FA affects morphological differences other than through influencing gene expression. I think this is general knowledge that for a molecule to cause phenotypic difference, it has to affect gene expression that results into the observable phenotypic differences. Rephrase please.
Response: We replaced “In addition to affecting the morphological aspects of development, the supply of FA can also influence the expression level of genes in the small intestine” with “The results of this trial showed that not only the morphological aspects of development were affected, the supply of FA can also influence the genes expression level in the small intestine” in line 314.
Line282-283: I barely understand what you are trying to communicate by this sentence. You should highlight the results before you discuss them, instead of discussing or reviewing several citation, and then bring in results. The link between this sentence and the preceding sentences is barely recognizable.
Response: We changed this content to “Interestingly, the expression of IGF-I in twin lambs was significantly lower than that in triplet lambs, but the result of the small intestinal phenotype was that twins developed better than triplets. Insulin-like growth factors (IGFs) is an important growth factor that regulates the development of small intestine [37]. Due to the role of IGF-I in promoting division and differentiation of cell, it has been found in pigs [38] and sheep [39] that IGF-I supplementation could improve the morphology of small intestine and promote the development of small intestinal epithelium” to highlight the results before discussion (line 316-319).
Line286-287: What do you mean by this “the fetal programming determined by epigenetic 286 modification can be adjusted by influencing the mRNA abundance” What modifies the mRNA abundance, without changing the epigenome. I do not think mRNA abundance can be influenced by not affecting epigenome first. Please this.
Response: We changed this content to “Comparing with twins, triplets compete more for maternal nutrition during fetal stage and are more likely to suffer from developmental defects caused by inadequate nutrition. The disturbance of environmental stress caused by undernutrition on ontogenetic development will reduce the adaptability of animals.” to make the logic clearer (line 321-324).
Line289-291: This explanation does not really sound fitting here, try to internalize your data before finding explanation for a few results you pick on, look at the large picture of your results. For example in your study you three levels of FA supplementation with one being zero supplementation, I would you to highlight what trends do you see when you supplement with 32mg than 16mg. Do mean that supplement 32mg also still allows for FA deficiency? Please spend some time reinternalizing your results as a whole not just bits. Could you discuss the observed differences between the studied two tissues, duodenum and the jejunum if any?
Response: We rewrote this content as “Therefore, a possible explanation for the higher expression of IGF-I in the jejunal tissue of triplets than twins is that the body compensates for the lower small intestine weight of triplet-born lambs by increasing expression of IGF-I. A similar experiment found that the intestinal morphology of offspring piglets with nutritional restriction during gestation was impaired, while FA supplementation increased the expression of anti-apoptotic genes in jejunum tissues [42]” in line 327-332.
Line292: Which previous results are you referring to here? Any citations?
Response: We deleted this sentence according to your suggestion.
Line293-294: Please write based on your results, this statement “the intestinal morphology of triplets was poorer than that of twins, resulting in high expression of development-related genes.” Is not true according to all the results you present in this paper. Please internalize your results and states where increases where observed and where decreases were observed and then discuss that.
Response: We reinterpreted our results, which is just a hypothesis, but we've seen similar findings in other studies. And we also reformulated the altered phenotypes or genes. The original content was changed to “Therefore, a possible explanation for the higher expression of IGF-I in the jejunal tissue of triplets than twins is that the body compensates for the lower small intestine weight of triplet-born lambs by increasing expression of IGF-I. A similar experiment found that the intestinal morphology of offspring piglets with nutritional restriction during gestation was impaired, while FA supplementation increased the expression of anti-apoptotic genes in jejunum tissues [42].” (line 327-332).
Line293-294: “the intestinal morphology of triplets was poorer than that of twins, resulting in high expression of development-related genes.” How can gene expression result from a phenotype? It is the other way round please think about a better discussion after internalizing your results.
Response: We made a possible hypothesis by combining phenotypic and genetic results, and we saw a similar explanation in another study. And we added “Therefore, a possible explanation for the higher expression of IGF-I in the jejunal tissue of triplets than twins is that the body compensates for the lower small intestine weight of triplet-born lambs by increasing expression of IGF-I. A similar experiment found that the intestinal morphology of offspring piglets with nutritional restriction during gestation was impaired, while FA supplementation increased the expression of anti-apoptotic genes in jejunum tissues [42].” (line 327-332).
Line295-296: “Gene expression can be reversed, but phenotypic reversal is very difficult.” Then? This sentence sounds redundant.
Response: We deleted this sentence according to your suggestion.
Line308-310: Please delete this “For example, apoptotic cells in the small intestine were significantly increased in celiac disease, suggesting that increased apoptotic cells were associated with villi atrophy in the disease [33]”
Response: We deleted this sentence according to your suggestion.
Line315-318: Provide a citation/citations for this information.
Response: The citation for this information is the 46th reference, and it is also citation for the next sentence, so it is marked at the end of the next sentence in line 351.
Line325-327: This statement is not true for all the cases, in some cases controls actually had higher expression of LCT than FA supplemented (see Triplets CO vs F32 duodenum) or (see Twins CON vs F16). For SGLT1 (see Triplets CO vis F16 or F32) and (Triplet CO vs F32).You could say that, “In this study, dietary FA supplementation in gestation generally increased expression of …………” Please review your results and discuss them as a whole, not picking a few.
Response: We added “generally” between “dietary FA supplementation in gestation” and “increased the expression” to make this statement conform to the result (line 356).
Line327-329: “which is consistent with what we found in terms of small intestine development-related gene expression.” What do you mean by this? Please rephrase this doesn’t add any new information.
Response: We rephrased it to “We have found that the genes expression of SI and SLC2A5 significantly increased in triplets compared with that in twins, which is consistent with the result of IGF-I expression in jejunum tissue” in line 361-363.
Line332-335: I still do not get the logic of your explanation, here you are talking in terms of absorption genes. Try to link your results together to get good explanation for your results. For example, in this case an increased expression of absorption related genes in triplets could actually be compensation for the small size (in terms of weight look at Table 4) absorptive surface area, since in your results you mention that triplet had smaller small intestines as compared to twins. In this case the animal would compensate for this reduced size. I again suggest you revisit your results and consider discussing them as a whole, you have very good results from which you can draw good discussion.
Response: According to your suggestion, we changed this content to “In this case, the increased expression of digestion-related genes in triplets could actually be compensation for the smaller absorptive surface area in the small intestine.” in line 364-366.
Conclusions
Line343-349: Please revise the conclusions giving general conclusions on the treatments you studied and what are the implications of your observations. You categorized the genes you studied, please general conclusion on those groups even if its just a trend, not just picking on just a few.
Response: We changed “the expression of IGF-I, BCL-2 and SGLT1, as well as reduce the expression of BAX in newborn lambs.” to “the expression of development-, antiapoptotic- and digestion-related genes, as well as reduce the expression of proapoptotic in newborn lambs.” according to your suggestion (line 380-381).

Reviewer 4 Report
The manuscript reports on the effect Maternal folic acid supplementation differently affects the small intestinal phenotype and gene expression of newborn lambs of different litter size. The manuscript is adequately developed in every part. The introduction is adequate and the materials and methods are described in a simple and understandable way. The results are accompanied by graphs and tables that highlight the results obtained. The discussion is too long and some parts can be cut because they are superfluous.
In my opinion, the manuscript can be published
Kind regards
Author Response
Response to Reviewer 4 Comments
Dear Reviewer,
Thank you for helping us improve the quality of manuscript (ID: animals-963445). We gratefully appreciate the constructive comments made by you. We revised the manuscript thoroughly according to your comments. Any changes of the manuscript were indicated in red in the revised manuscript and listed as follows.
Reviewer 4
Comments and Suggestions for Authors
The manuscript reports on the effect Maternal folic acid supplementation differently affects the small intestinal phenotype and gene expression of newborn lambs of different litter size. The manuscript is adequately developed in every part. The introduction is adequate and the materials and methods are described in a simple and understandable way. The results are accompanied by graphs and tables that highlight the results obtained. The discussion is too long and some parts can be cut because they are superfluous.
In my opinion, the manuscript can be published
Kind regards
Response: According to your suggestion, we modified the discussion section to make it more concise. Thanks very much for your constructive comments and recognition, we will continue to work hard.

Round 2
Reviewer 3 Report
I am satisfied by the response of the authors to my suggestions and questions and I recommend acceptance of the manuscript.
Thank you
Author Response
Thank you for your time.